# BEVCA: Effective and Transferable Camou­flage Attack against Multi-view BEV-based 3D Perception in Autonomous Driving

## Abstract

Multi-view 3D perception models are widely adopted by leading car manufactur­ers due to their highly competitive performance. However, existing adversarial camouflage techniques primarily focus on single-view 2D detectors, limiting their effectiveness against multi-view 3D perception models. In the paper, we propose BEVCA, the first framework to generate adversarial camouflage that effectively attacks multi-view BEV-based 3D perception models by exploiting the Bird's-Eye-View (BEV) representation used across various 3D perception models. Our framework introduces a new differentiable multi-view neural renderer to enable end-to-end gradient-based camouflage optimization. Furthermore, we propose a novel BEV-feature-based adversarial loss to achieve effective and transferable at­tacks. Extensive experiments on 3D object detection and segmentation scenarios demonstrate that BEVCA outperforms the best existing baselines, achieving aver­age attack improvements of 36.2% and 21.6% in black-box settings, respectively. Our code is available at https://anonymous.4open.science/r/BEVCA-1D82.

## 1 Introduction

Vision-based autonomous driving (AD) systems have been widely adopted by major vehicle man­ufacturers, such as Tesla Tesla, Inc. (2025), due to their competitive performance. These systems utilize multiple camera sensors surrounding the ego vehicle to perceive the nearby environment with deep learning models Philion & Fidler (2020); Zhou & Krähenbühl (2022); Runsheng Xu (2022); Huang et al. (2021); Wang et al. (2022b); Li et al. (2022). However, despite the significant progress, the current research shows deep neural networks (DNNs) are proven to be vulnerable to adversarial examples Szegedy et al. (2014), which threaten the safety-critical nature of the AD systems.

Prior attacks mainly focus on the single-view 2D vehicle detection task. These methods Wang et al. (2022a); Suryanto et al. (2022; 2023); Zhou et al. (2024a); Lyu et al. (2024); Zhou et al. (2024b) use the Carla simulator Dosovitskiy et al. (2017) to generate vehicle images with various transformations to enhance the attack robustness. Besides, they leverage a differentiable neural renderer Kato et al. (2018); Ravi et al. (2020) to enable direct adversarial camouflage optimization against the target model, achieving effective attack performance.

Despite the impressive progress made on 2D attacks, the current state-of-the-art vehicle detection methods are dominated by 3D perception models, which take multi-view images as input. As a result, the current single-view 2D attack pipelines cannot be directly applied to the multi-view 3D perception models. Some works like Zhu et al. (2023); Wang et al. (2025) explore attacking the multi-view 3D perception models with adversarial patches by minimizing the detection scores. However, the adversarial patches are inherently not robust against various viewing angles since they cannot fully cover the target vehicle, leading to suboptimal attack performance. Besides, attacks in the real world are mostly black-box settings, yet all the previous methods require access to the target model's detection scores to optimize the adversarial texture, which limits their attack transferability.

To tackle the above challenges, we propose BEVCA, a novel framework to generate adversarial camouflage against multi-view BEV-based 3D perception models. The insight of our framework is: most of the state-of-the-art multi-view 3D perception models Philion & Fidler (2020); Huang et al. (2021); Wang et al. (2022b); Li et al. (2022) rely on the bird's-eye-view (BEV) feature extracted

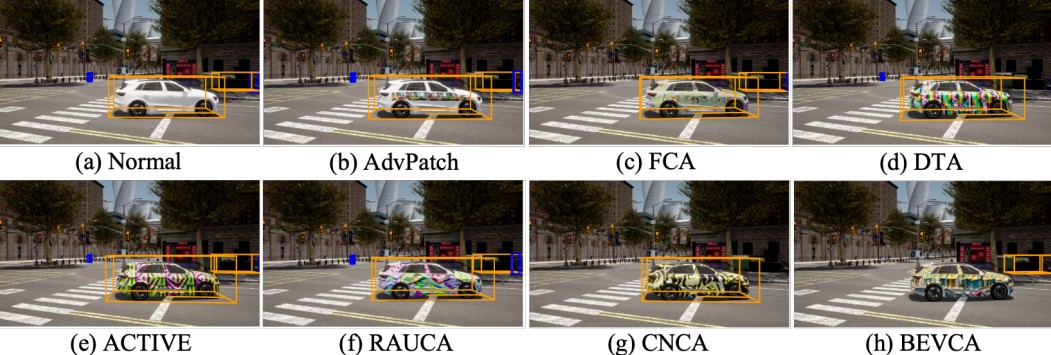

Figure 1: Comparison of 3D detection results using different textures on the photo-realistic simulation environment, where only the adversarial camouflage generated by our BEVCA framework successfully evades detection. (a) A normal vehicle without attack texture; (b) AdvPatch Wang et al. (2025). (c)-(g) are the top-performing methods for 2D detection: FCA Wang et al. (2022a), DTA Suryanto et al. (2023), ACTIVE Suryanto et al. (2023), RAUCA Zhou et al. (2024a), and CNCA Lyu et al. (2024) respectively; (h ) Our method BEVCA.

from the multi-view input, which is a versatile 2D feature map that can be used for various perception tasks in AD systems Hu et al. (2023); Zheng et al. (2024). Therefore, the attack on the BEV feature can result in attacks on the downstream tasks. Motivated by this insight, we first enable the neural rendering of multi-view, realistic, and consistent target vehicle images, facilitating direct gradient-based camouflage generation against multi-view 3D perception models. After that, we introduce a novel adversarial attack based on the BEV feature. We precisely locate the BEV feature regions that are highly associated with the target vehicle to compute the adversarial loss. As a result, the combination of the multi-view neural rendering and BEV-feature-based adversarial attack results in highly effective and transferable adversarial camouflage for different perception tasks and models.

The main contributions of our work are summarized as follows:

- To the best of our knowledge, our framework is the first to generate adversarial camouflage against multi-view BEV-based 3D perception models. It enables effective and transferable adversarial attack via novel multi-view neural rendering and the BEV-feature-based adversarial attack.

- We propose a novel neural renderer capable of generating realistic and multi-view target vehicle images from different positions to the ego vehicle, facilitating gradient-based adversarial camouflage optimization against multi-view BEV-based 3D perception models.

- We propose a novel adversarial loss function to attack in the BEV feature space associated with the target vehicle, which leads to effective and transferable attacks against various BEV-based 3D perception models and tasks.

We conduct a comprehensive evaluation with various multi-view BEV-based 3D perception models. The results show that our generated adversarial camouflage outperforms existing baselines, achieving average improvements of 36.2% and 21.6% in 3D object detection and segmentation, respectively.

## 2 RELATED WORK

In this section, we first introduce multi-view 3D perception, then review adversarial camouflage attacks on vehicle detection, and finally define the threat model for our scenario.

### 2.1 MULTI-VIEW 3D PERCEPTION

The development of large-scale multi-view datasets such as NuScenes Caesar et al. (2020) and Waymo Open Dataset Sun et al. (2020) has significantly accelerated progress in multi-view 3D per-

ception. Bird's-Eye-View (BEV) models Philion & Fidler (2020); Huang et al. (2021); Wang et al. (2022b); Li et al. (2022) have demonstrated state-of-the-art performance across various autonomous driving (AD) tasks. These models extract and aggregate information from multi-view driving images to construct a unified BEV feature representation of the environment, which can support various downstream tasks, including 3D object detection, map segmentation, and even end-to-end driving systems Hu et al. (2023); Zheng et al. (2024).

## 2.2 ADVERSARIAL CAMOUFLAGE

Accurate vehicle detection is critical for the safety of AD systems, motivating a growing interest in studying the robustness of vehicle detectors with adversarial attacks. These attacks can be categorized as single-view 2D attacks and multi-view 3D attacks. Existing 2D attack methods leverage 3D simulators such as Carla Dosovitskiy et al. (2017) to generate target vehicle images under various transformations for optimizing adversarial textures. Then, these methods Wang et al. (2022a); Suryanto et al. (2022; 2023); Zhou et al. (2024a); Lyu et al. (2024) employ the differentiable neural renderer Kato et al. (2018); Ravi et al. (2020) to directly optimize adversarial textures via gradient backpropagation. Besides, Suryanto et al. (2022; 2023); Zhou et al. (2024a); Lyu et al. (2024) further incorporate realistic environmental features, such as shadows and fog, to mitigate the domain gap between simulation and real-world conditions, achieving better attack performance. Regarding multi-view 3D attacks, previous works attempt to generate the adversarial patches on the target vehicle. Zhu et al. (2023); Wang et al. (2025) utilize the 3D annotations of the target vehicle to transform the 2D adversarial patches so that they look realistic and consistent in the multi-view image setting, while Li et al. (2024) utilizes NeRF to generate adversarial texture on the vehicle.

In contrast to prior works, our framework introduces a novel combination of multi-view neural rendering and BEV-feature-based adversarial attack to enable effective and transferable attacks on multi-view BEV-based 3D perception models.

## 2.3 THREAT MODEL

Our attack framework targets multi-view BEV-based 3D perception models in AD systems, aiming to craft physically realizable camouflage attacks that degrade perception performance on a targeted vehicle.

**Targeted Model:** We consider attackers targeting multi-view perception models that construct a BEV feature representation for downstream tasks like object detection and segmentation. These BEV perception models, for instance, BEVFormer Li et al. (2022), are widely adopted in the current state-of-the-art multi-view 3D perception tasks Hu et al. (2023); Zheng et al. (2024).

**Attacker:** The attacker's goal is to reduce the perception performance of the target models towards the camouflaged vehicle, subject to the physical constraints and consistently effective across various viewpoints. This is achieved by altering the vehicle's appearance with adversarial camouflage textures. In the white-box setting, the attacker is assumed to have full access to the targeted model for gradient-based optimization. In the black-box setting, the attacker has no internal access to the targeted model but knows the targeted model relies on BEV feature representation. In both cases, the attacker cannot modify training data, model weights, or sensor configurations.

## 3 METHODS

In this section, we first present an overview of our framework for generating effective and transferable adversarial camouflage against multi-view 3D perception models. Then, we describe the details of the essential components of our framework.

## 3.1 OVERVIEW

Figure 2 illustrates the overall framework for generating adversarial camouflage. We first construct a multi-view image dataset of a target vehicle captured from the ego vehicle's perspective using the Carla simulator Dosovitskiy et al. (2017), following the NuScenes Caesar et al. (2020) dataset format. This dataset contains the multi-view images $X_{in}$, camera configurations $\Phi_{cam}$, and the

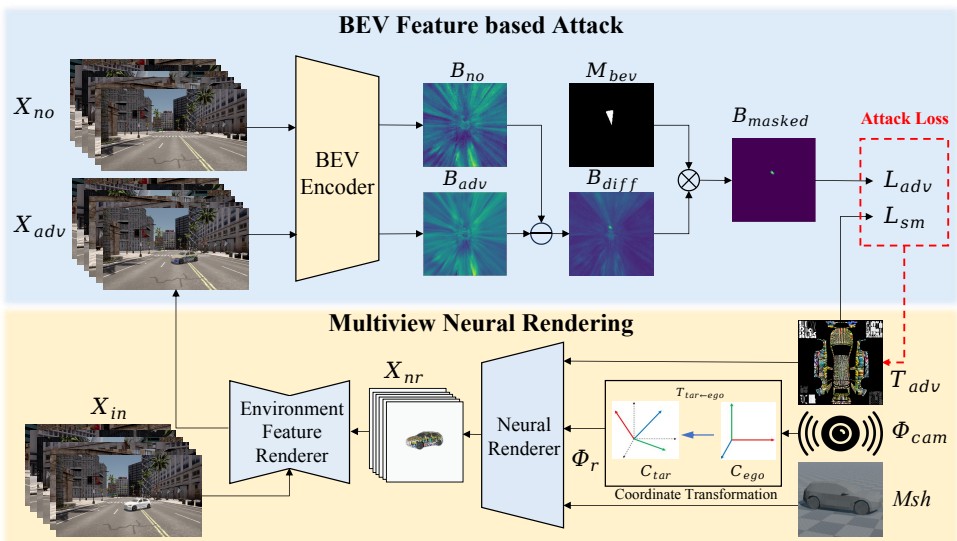

Figure 2: The overview of BEVCA. Our framework consists of multi-view neural rendering and BEV-feature-based attack modules. The multi-view neural rendering outputs the multi-view camouflage vehicle images from the ego vehicle perspective. The BEV-feature-based attack locates the BEV feature regions that are highly associated with the target vehicle with a designed mask. The combination of both modules enables effective and transferable camouflage attacks against multi-view 3D perception models.

transformation matrices $T_{tar \leftarrow ego}$ that map ego coordinates to target vehicle coordinates. Our multi-view neural renderer first uses a neural renderer $NR$, which takes the adversarial texture $T_{adv}$, 3D vehicle mesh $Msh$, camera configurations $\Phi_{cam}$, and the transformation matrices $T_{tar \leftarrow ego}$ as input, and produces the multi-view rendered adversarial target vehicle images $X_{nr}$. Subsequently, both $X_{nr}$ and $X_{in}$ are input into an Environmental Feature Renderer (EFR). We use the pretrained EFR from Lyu et al. (2024) to extract environmental features from $X_{in}$ and render them together with the background to output the realistic multi-view camouflaged target vehicle images $X_{adv}$. To summarize:

$$X_{adv} = EFR(X_{in}, NR(Msh, T_{adv}, \Phi_{cam}, T_{tar \leftarrow ego})) \qquad (1)$$

Instead of attacking task-specific output scores (e.g., detection confidence), we target the BEV feature representation to improve attack transferability. A key challenge is identifying the regions within the BEV feature map that correspond to the target vehicle's presence in the multi-view images. To this end, we construct a *no-vehicle* multi-view dataset $X_{no}$ under the identical settings as $X_{in}$, except that the target vehicle is removed. We feed $X_{adv}$ and $X_{no}$ into a BEV encoder to obtain the corresponding BEV features:

$$B_{adv} = BEV(X_{adv}) \qquad (2)$$
$$B_{no} = BEV(X_{no}) \qquad (3)$$

Since the only difference between $X_{adv}$ and $X_{no}$ is the presence of the target vehicle, we hypothesize that the difference between $B_{adv}$ and $B_{no}$ reflects the target vehicle's impact on BEV feature space:

$$B_{diff} = |B_{adv} - B_{no}| \qquad (4)$$

Empirically, we observe that the prominent regions in $B_{adv}$ are geometrically aligned with the target vehicle's ground-truth location. Based on this observation, we apply a BEV mask $M_{bev}$ to $B_{diff}$, locating the affected regions $B_{masked}$:

$$B_{masked} = B_{diff} \cdot M_{bev} \qquad (5)$$

We define an adversarial loss $L_{adv}$ based on $B_{masked}$ to attack the BEV representation towards the target vehicle. Additionally, following Sharif et al. (2016), we employ a total variation loss $L_{tv}$ on the adversarial texture image $T_{adv}$ to ensure the naturalness of texture to human vision. The details of $L_{adv}$ and $L_{tv}$ will be discussed in the following section. Finally, our total loss function to optimize the adversarial texture is:

$$L_{total} = \alpha L_{adv} + \beta L_{tv}, \tag{6}$$

where $\alpha$, $\beta$ are the hyperparameters to control the contribution of each loss function.

## 3.2 MULTI-VIEW NEURAL RENDERING

Previous works Suryanto et al. (2022; 2023); Zhou et al. (2024a); Lyu et al. (2024) have demonstrated the effectiveness of adversarial camouflage against 2D object detectors by leveraging realistic neural rendering of textured vehicles. In the single-view setting, the adversarial vehicle is placed at the center of the image, and the rendering can be achieved using the object-centric rendering methods from Ravi et al. (2020); Kato et al. (2018). These methods only require the relative camera pose with respect to the adversarial vehicle, which can be easily obtained during the camera initialization in CARLA. However, this pipeline is not directly applicable to multi-view 3D perception tasks. In such scenarios, multiple cameras are attached to the ego vehicle to observe the target vehicle. As a result, the camera pose relative to the target vehicle—required by the renderer—can no longer be retrieved through the simple initialization procedure used in the single-view pipeline. To support multi-view rendering, it is necessary to redesign the pipeline by computing the relative pose between each ego-mounted camera and the target vehicle, based on the geometric relationships among the ego vehicle, the target vehicle, and the cameras.

To address this, we extend the popular differentiable rendering library PyTorch3D Ravi et al. (2020) as the naive renderer to support multi-view rendering. The naive renderer requires three key inputs: the 3D mesh of the vehicle, the adversarial texture, and the camera settings relative to the target vehicle. The key challenge is to compute the transformations $T_{tar \leftarrow ego}$, which convert the camera settings of each camera coordinate system $\Phi_{cam}$ to the target vehicle's coordinate system:

$$\Phi_r = T_{tar \leftarrow ego} \cdot \Phi_{cam} \tag{7}$$

where $\Phi_r$ can be directly input to the naive neural renderer. These transformations ensure that each rendered view accurately reflects the camera's perspective towards the target vehicle. We apply these camera parameter transformations to all six ego-mounted cameras and perform image rendering as defined in Eq 1 to obtain multi-view rendering images. The full mathematical details and transformation steps are provided in the appendix A.1 for interested readers.

## 3.3 BEV-FEATURE-BASED ADVERSARIAL ATTACK

The current state-of-the-art multi-view 3D perception models Philion & Fidler (2020); Huang et al. (2021); Wang et al. (2022b); Li et al. (2022) rely heavily on BEV features to support downstream perception tasks in autonomous driving systems Hu et al. (2023); Zheng et al. (2024). This motivated us to attack from the BEV feature to achieve an effective and transferable attack. However, since the BEV feature is high-dimensional, it is challenging to directly isolate the feature areas that are influenced by the target vehicle. To tackle this, we utilize the BEV feature extracted from the dataset with the same camera configurations but excluding the target vehicle. We analyze the difference between these two BEV features to locate the target vehicle's associated feature regions, which we used to calculate the adversarial loss.

Specifically, as shown in Eq. 2 and 3, we first obtain the BEV features $B_{adv}$ and $B_{no}$ from $X_{adv}$ and $X_{no}$ respectively. We compute their difference to obtain the $B_{diff}$, which has the shape of $(H, W, C)$, where $H$ and $W$ define the BEV perception grid, and $C$ is the feature dimension. To localize the affected regions, we average across the feature dimension C and visualize the resulting 2D heatmap. Empirically, as shown in Figure 3, we observe that high-activation areas in this map tend to coincide with the ground-truth location of the target vehicle, suggesting a strong correlation.

Based on the above observation, we propose a BEV mask $\mathcal{M}_{bev}$ to effectively crop out the affected regions. We leverage the annotated position of the target vehicle to build this mask. For convenience,

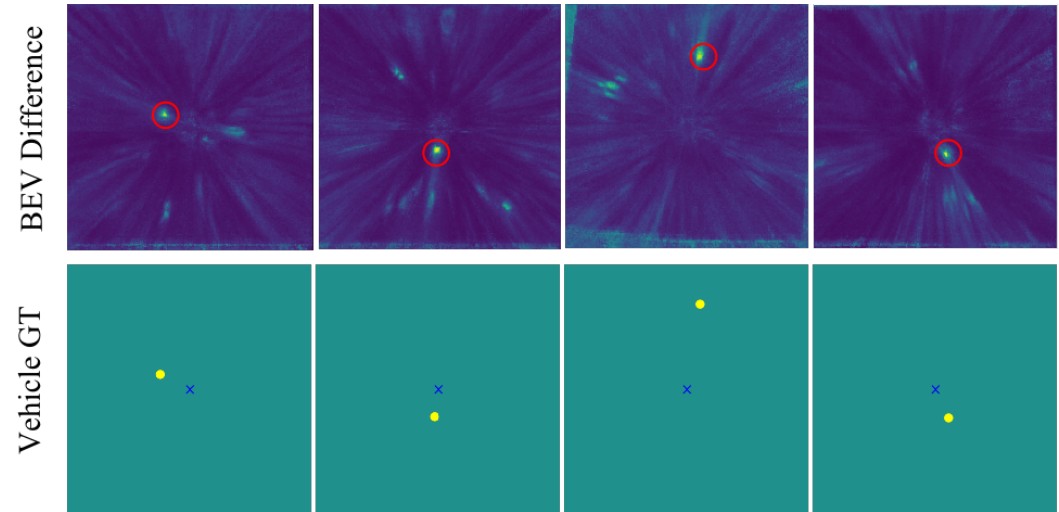

Figure 3: Examples of BEV difference feature and corresponding vehicle ground truth images. We can observe a strong correlation between BEV regions relevant to the target vehicle (marked by the red circles) and vehicle ground truth position (marked by the yellow points).

we represent the target's position in polar coordinates $(r_{tar}, \theta_{tar})$ related to the ego vehicle and define the mask region as:

$$\mathcal{M}_{bev} = \{(r, \theta) \,|\, |r - r_{\text{tar}}| \leq \Delta r, \ |\theta - \theta_{\text{tar}}| \leq \Delta \theta\} \tag{8}$$

where $\Delta r$ and $\Delta \theta$ are the tunable thresholds to control the mask area. In the ablation section, we evaluate the effectiveness of this mask strategy.

Once the masked BEV difference $B_{masked}$ is obtained (as shown in Eq. 5. ), we define the adversarial $L_{adv}$ as the mean of the non-zero elements in $B_{masked}$:

$$L_{adv} = \frac{\sum\limits_{i} B_{masked,i}}{\sum\limits_{i} \Bbbk(B_{masked,i} \neq 0)} \tag{9}$$

where $\Bbbk(\cdot)$ is the indicator function, which returns 1 if the condition inside is true and 0 otherwise. The denominator represents the number of non-zero elements in $B_{masked}$, and the numerator is the sum of all elements in $B_{masked}$.

Additionally, we also apply the Total Variation (TV) loss followed by Sharif et al. (2016) to encourage naturalness of the texture towards human vision:

$$L_{tv} = \sum_{i,j} \left(x_{i,j} - x_{i+1,j}\right)^2 + \left(x_{i,j} - x_{i,j+1}\right)^2 \tag{10}$$

## 4 EXPERIMENTS

### 4.1 EXPERIMENTAL SETTINGS

**Datasets.** We utilize the Carla simulator Dosovitskiy et al. (2017) to generate datasets for our experiments. To have a comparative analysis with prior studies Wang et al. (2022a); Suryanto et al. (2023), we select the Audi E-Tron as the target vehicle model. We generate a dataset of 2500 samples for camouflage generation and a test dataset of 500 samples for camouflage evaluation. These datasets contain multi-view images of the target vehicle with various distances and view angles relative to the ego vehicle, across various background scenes. During our experiments, we found that the performance of the original targeted models degraded significantly in the simulation

dataset compared to their reported results, as shown in Table 7. This is because these models are trained on realistic datasets like nuScenes Caesar et al. (2020). To alleviate the domain gap between simulation and realism, we generate a fine-tuning dataset of 500 samples with completely different settings from camouflage generation and test datasets to adapt these models to the digital domain. For the real-world evaluation, we print our camouflage in a 2D UV map image and stick it to a full-size Audi vehicle. We build a simple multi-view camera data collection platform to collect real-life multiview data. We collect two datasets: one of the car with normal painting and another of the car with our camouflage. Each dataset has 191 samples with different azimuths and distances towards the ego position. We split the dataset into close (less than 10 meters) and long scenarios(more than 10 meters) based on the distance between target vehicle and ego position.

**Baselines.** We compare our method with state-of-the-art solutions: advanced adversarial camouflages against 2D detection and adversarial patches against 3D detection: FCA Wang et al. (2022a), DTA Suryanto et al. (2022), ACTIVE Suryanto et al. (2023), RAUCA Zhou et al. (2024a), CNCA Lyu et al. (2024), and AdvPatch Wang et al. (2025). For a fair comparison, we use the official textures generated by these methods and apply them to the Audi E-Tron model.

**Evaluation metrics.** In digital experiment, to evaluate the effectiveness of the adversarial camouflage, we utilize the Average Precision (AP) score defined by the 2D center distance between the prediction and ground truth on the ground plane Caesar et al. (2020); Li et al. (2022) for 3D detection task and Intersection-over-Union (IoU) Zhou & Krähenbühl (2022) of vehicle class for BEV segmentation task, respectively. In the real-world experiment, we use the Attack Success Rate (ASR) which measures the percentage at which the target vehicle is successfully detected originally but not detected after the attack.

**Target models.** We choose the BEVFormer-base Li et al. (2022) model as the white-box target model for adversarial camouflage generation. To evaluate the effectiveness and transferability of the optimized camouflage, we utilize a collection of 3D object detection and BEV segmentation models treated as black-box models. 3D object detection models include BEVFormer series (small, tiny, base-seg-det, small-seg-det) Li et al. (2022) with different model configurations. The BEV segmentation models include Lift-Splat-Shoot Philion & Fidler (2020), Cross View Transformer Zhou & Krähenbühl (2022) and SinBEVT Runsheng Xu (2022). We train the original version of these models on the fine-tuning dataset with one epoch before evaluating on the test dataset.

**Optimization details.** We utilize the Adam optimizer with a learning rate of 0.01 for adversarial camouflage generation. The camouflage texture is initialized randomly and trained with 10 epochs. For the hyperparameters of loss functions, we set the values of $\alpha$ and $\beta$ (see Eq. 6) to 1 and 1000, respectively 5. For the BEV mask, we set the distance and angle thresholds ($\Delta r$, $\Delta \theta$) to 15 meters and 15 degrees (see Eq. 8) after hyperparameter tunings 6. We conduct experiments on one NVIDIA RTX A800 80GB GPU.

## 4.2 ATTACK PERFORMANCE EVALUATION

**Attack performance on 3D perception models.** We compare our proposed method, BEVCA, against several state-of-the-art adversarial camouflage approaches, including FCA, DTA, ACTIVE, RAUCA, CNCA, and AdvPatch. To thoroughly evaluate the effectiveness and transferability of our method, we conduct extensive experiments across a variety of multi-view 3D perception models in both 3D object detection and BEV segmentation tasks. The results are shown in Table 1, where BEVCA consistently outperforms all baselines. AdvPatch shows performance comparable to that of a normally painted vehicle, primarily due to its reliance on small, localized patches. FCA provides only marginal improvement, as it lacks the ability to model complex environmental interactions. In contrast, DTA, ACTIVE, RAUCA, and CNCA offer stronger performance, yet remain suboptimal because they are not explicitly designed for multi-view perception models. Our method surpasses all the baselines, achieving an average performance gain of approximately 36.2% in black-box settings against the best previous baseline, demonstrating strong effectiveness and transferability.

We also provide some adversarial camouflage vehicle examples against 3D detection in different scenarios in the appendix section . These examples demonstrate that the vehicle with normal painting is accurately detected with well-aligned 3D bounding boxes. However, with our adversarial camouflage texture, the target vehicle becomes significantly harder to detect, often resulting in inaccurate bounding boxes or complete disappearance from the detection output.

Table 1: Comparison of the effectiveness of attacks across various 3D detection and BEV segmentation models. The metrics for 3D detection and BEV segmentation are AP and IoU of the vehicle class, respectively.

| METHODS | 3D DETECTION | | | | | BEV SEGMENTATION | | |
|---|---|---|---|---|---|---|---|---|
| | BASE | SMALL | TINY | BASE-SEG-DET | SMALL-SEG-DET | LSS | CVT | SINBEVT |
| NORMAL | 0.618 | 0.721 | 0.514 | 0.750 | 0.634 | 0.509 | 0.609 | 0.570 |
| FCA | 0.593 | 0.696 | 0.512 | 0.721 | 0.595 | 0.442 | 0.505 | 0.489 |
| DTA | 0.578 | 0.663 | 0.469 | 0.684 | 0.591 | 0.372 | 0.480 | 0.449 |
| ACTIVE | 0.559 | 0.664 | 0.433 | 0.682 | 0.588 | 0.367 | 0.481 | 0.436 |
| RAUCA | 0.562 | 0.682 | 0.479 | 0.689 | 0.576 | 0.380 | 0.459 | 0.427 |
| CNCA | 0.567 | 0.626 | 0.404 | 0.634 | 0.523 | 0.306 | 0.401 | 0.360 |
| ADVPATCH | 0.601 | 0.702 | 0.516 | 0.703 | 0.608 | 0.479 | 0.579 | 0.533 |
| BEVCA | **0.191** | **0.285** | **0.381** | **0.329** | **0.331** | **0.253** | **0.307** | **0.273** |

For the BEV segmentation task, although the adversarial camouflage generated by BEVCA is specifically optimized for the BEVFormer base model, it consistently achieves the best attack performance across all segmentation models when compared to existing baselines. Notably, BEVCA attains an average improvement of approximately 21.6% over the strongest previous baseline performance in black-box settings. These results further underscore the task transferability and robust generalization of our approach across diverse BEV-based perception tasks.

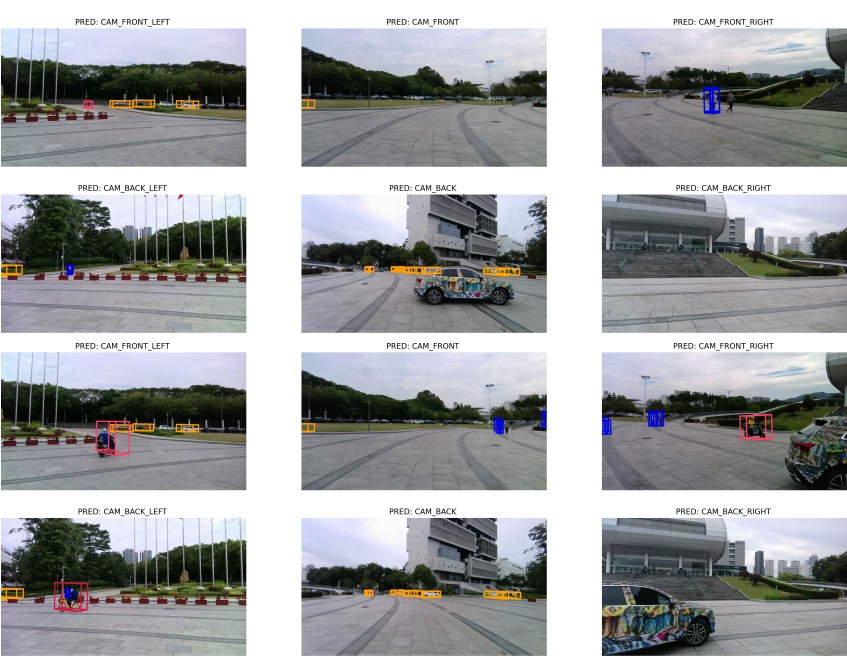

Figure 4: Visualization results of our camouflage attack against 3D object detection in the physical world.

**Multi-view robust attack.** To evaluate the robustness of our adversarial camouflage under varying target vehicle positions, we conduct comprehensive experiments across different distances and azimuths relative to the ego vehicle. Specifically, we generate a dataset where the target vehicle appears at distances of [10, 15, 25, 40] meters, aligned with the BEVFormer model's perception range of up to 51.2 meters. For azimuths, we sample [0°, 60°, 120°, 180°, 240°, 300°]. For each distance-azimuth combination, we create 40 samples with varying background scenes across different CARLA towns to ensure diversity. We use the BEVFormer base model for evaluation, as other detectors exhibit similar trends.

Table 2: The attack performance for multi-azimuth and multi-distance. Notice the AP of the normal painted car is 0.618.

| AZIMUTH (°) | DISTANCE | | | | AVG |
|---|---|---|---|---|---|
| | 10 | 15 | 25 | 40 | |
| 0 | 0.278 | 0.129 | 0.468 | 0.376 | 0.313 |
| 60 | 0.005 | 0.184 | 0.311 | 0.278 | 0.195 |
| 120 | 0.206 | 0.332 | 0.167 | 0.218 | 0.231 |
| 180 | 0.123 | 0.641 | 0.404 | 0.001 | 0.292 |
| 240 | 0.242 | 0.293 | 0.300 | 0.195 | 0.258 |
| 300 | 0.088 | 0.282 | 0.347 | 0.197 | 0.228 |
| AVG | 0.157 | 0.310 | 0.333 | 0.211 | 0.253 |

Table 3: Attack success rate of our camouflaged car in the physical world evaluation

| Distance | Close | Long |
|---|---|---|
| ASR | 53.1% (51/96) | 15.8% (15/95) |

The results are shown in Table 2. The total mean Car AP value is comparable to the value reported in Table 1. We also observe that with the increase of the distance, the AP first prone to increase at a distance of 10, after that the AP prone to decrease after a distance of 25. This trend can be attributed to two factors: (1) with the increase of the distance, fine-grained adversarial camouflage patterns are difficult for the detection model to identify, causing the decrease of the attack performance; and (2) the detection performance of the model declines at long ranges, causing the AP to decrease again. Nevertheless, our camouflage maintains strong attack effectiveness across all tested configurations, demonstrating its robustness in multi-view settings.

**Physical world evaluation.** We conduct the physical world evaluation by printing the camouflage in a 2D UV map format and sticking it to a full-size Audi car. To collect real multi-view data, we built a simple multi-view camera data collection platform following the camera setup from the nuScenes Caesar et al. (2020) dataset. We collect two datasets: one of the car with normal painting and another of the car with our camouflage. Then we compare the detection results of these two datasets with BEVFormer-base model to compute ASR scores. Table 3 shows the attack performance of our method in both close and long distance settings. Our method achieves a high attack rate of 53.1% in close distance scenarios, demonstrating a realistic threat in real-world applications. Furthermore, Figure 4 shows that our camouflaged car can be successfully undetected from the 3D detector while the other surrounding objects (cars, scooters, pedestrians) are well detected. In summary, the physical evaluation results demonstrate that our method is transferable to the real world.

## 4.3 ABLATION STUDIES

Table 4: Comparison of the attack performance of different adversarial loss functions across various 3D detection and BEV segmentation models.

| METHODS | 3D DETECTION | | | | | BEV SEGMENTATION | | |
|---|---|---|---|---|---|---|---|---|
| | BASE | SMALL | TINY | BASE-SEG-DET | SMALL-SEG-DET | LSS | CVT | SINBEVT |
| $L_{det}$ | 0.564 | 0.683 | 0.498 | 0.728 | 0.597 | 0.450 | 0.505 | 0.467 |
| $L_{bev}$+ NO MASK | 0.612 | 0.704 | 0.508 | 0.735 | 0.585 | 0.498 | 0.588 | 0.574 |
| $L_{bev} + \mathcal{M}_{bev}$ | **0.191** | **0.285** | **0.381** | **0.329** | **0.329** | **0.253** | **0.307** | **0.258** |

**Effectiveness of different adversarial losses.** During the camouflage generation, we choose to optimize the camouflage with adversarial loss based on BEV features rather than downstream task detection scores. Furthermore, we propose a BEV mask to precisely locate the BEV feature regions that are highly associated with the target vehicle's influence. In these ablation studies, we compare the attack performance of the resulting camouflage of different adversarial losses to validate our

Table 5: Comparison of the attack performance of different total variation loss configurations.

| $\beta$ | BEVFORMER | | | AVG |
|---|---|---|---|---|
| | BASE | SMALL | TINY | |
| 0 | 0.537 | 0.689 | 0.488 | 0.571 |
| 10 | 0.453 | 0.630 | 0.457 | 0.513 |
| 100 | 0.358 | 0.553 | 0.425 | 0.445 |
| 1000 | **0.191** | 0.285 | 0.381 | **0.286** |
| 5000 | 0.199 | 0.286 | 0.374 | **0.286** |
| 10000 | 0.211 | **0.277** | **0.370** | **0.286** |

Table 6: The attack performance of different mask parameter settings.

| $\Delta r$ (°) | $\Delta\theta$ | | |
|---|---|---|---|
| | 5 | 15 | 25 |
| 5 | 0.209 | 0.221 | 0.205 |
| 15 | 0.216 | **0.191** | 0.228 |
| 25 | 0.219 | 0.210 | 0.223 |

proposed methods. We generate the adversarial camouflages with the following adversarial loss settings against the same target model (BEVFormer base): $L_{det}$ based on 3D object detection scores, $L_{bev}$ based on BEV features but without applying BEV masks, $L_{bev}$ with mask $\mathcal{M}_{bev}$.

We evaluate the camouflage attack performance across different models in both 3D object detection and BEV segmentation tasks in Table 4, respectively. The camouflage generated by $L_{det}$ achieves similar attack performance to baselines like FCA, demonstrating its limitation to transfer to other models and tasks. The camouflage generated by $L_{bev}$ without BEV mask show similar performance as the Normal baseline without any attack in both 3D detection and BEV segmentation tasks. This is because when using $L_{bev}$ with no mask as the adversarial loss, the corresponding attack input space is the entire multi-view image input. Using such adversarial loss to guide the optimization of the target vehicle texture is ineffective because the target vehicle area only occupies a very small part of the entire BEV perception area (approximately 3%), as shown in Figure 3. Therefore, we need to enhance the $L_{bev}$ with the mask to accurately locate the BEV feature areas that are highly relevant to the target vehicle. In our experiment, the camouflage generated by $L_{bev}$ with the BEV mask achieves the best performance in both 3D detection and BEV segmentation tasks across different models, demonstrating the effectiveness and transferability of our proposed pipeline.

**Effectiveness of hyperparameters during optimization.** We investigate the impact of the main hyperparameters during camouflage optimization related to total variation loss and BEV mask. Specifically, we vary the hyperparameter $\beta$ that controls the strength of the total variation constraint, and report the results in Table 5 with BEVFormer series models. The results show that incorporating the total variation loss consistently improves attack performance. When $\beta$ is too small, the generated textures lack coherent structure and fail to deliver strong adversarial signals—especially at long ranges. As $\beta$ increases, the attack performance improves steadily, highlighting the importance of enforcing total variation in the camouflage pattern. For BEV mask settings, we run hyperparameter tuning experiments with the BEVFormer-base model on two tunable thresholds $\Delta r$ and $\Delta\theta$ to find suitable values such that it can locate the relevant area accurately. The results from Table 6 show that $\Delta r$ with value of 15 and $\Delta\theta$ with values of 15 give the best results. We fix these settings throughout our experiments.

## 5 CONCLUSION & LIMITATION

We have proposed BEVCA, a novel adversarial camouflage attack framework against multi-view 3D perception. Our framework can generate effective and transferable adversarial camouflage for different 3D perception tasks and models. In particular, we propose a novel multi-view neural renderer to facilitate the gradient-based camouflage optimization against multi-view 3D perception models. Besides, we propose a novel adversarial loss based on BEV features to enable effective and transferable attacks. With extensive experiments against various black-box models in both 3D object detection and BEV segmentation tasks, the results demonstrate that BEVCA outperforms the existing works under multi-view 3D perception settings. Our current work is limited by the fact that we have not conducted more comprehensive physical experiments to evaluate the transferability of our method due to limited time and resources, which will be addressed in future work.

## 6 ETHICS STATEMENT

This paper presents work whose goal is to advance the safety of AD systems. While the proposed adversarial attack method could be potentially used by malicious users, it can also support efforts to enhance the robustness of AD systems via adversarial training, adversarial testing, and adversarial detection, thereby safeguarding the security of AD systems.

## 7 REPRODUCIBILITY STATEMENT

We provide the necessary materials in an open-source repository for readers who want to reproduce our work https://anonymous.4open.science/r/BEVCA-1D82. We provide our code and environment setup steps for camouflage generation. Besides, we also provide an optimized adversarial texture of our method for testing attack performance. Lastly, We will also upload the datasets for camouflage generation and testing, and the optimization details mentioned in the section 4.1.

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

# A  APPENDIX

## A.1  MULTI-VIEW CAMERA POSE TRANSFORMATION DETAILS

The key challenge for precise multi-view rendering of the target vehicle is to compute the transformation from each camera coordinate system to the target vehicle's coordinate system. First, we can obtain the following geometric data from the CARLA API:

- The global position and rotation of the ego vehicle: $t_{ego}^{glob}$, $R_{ego}^{glob}$.
- The global position and rotation of the target vehicle: $t_{tar}^{glob}$, $R_{tar}^{glob}$.
- The position and rotation of the ego camera in the ego vehicle frame: $t_{cam}^{ego}$, $R_{cam}^{ego}$.

Since CARLA employs a left-handed coordinate system while PyTorch3D uses a right-handed one, we first convert all relevant data accordingly through a proper handedness transformation.

$$\hat{t} = M \cdot t, \ \hat{R} = M \cdot R \cdot M, \ \text{where } M = diag(1, -1, 1) \tag{11}$$

where $\hat{t}$ and $\hat{R}$ denote transformed quantities in the right-handed coordinate system. With the above information, we can construct the following transformation matrices:

$$T_{tar \leftarrow glob} = \begin{bmatrix} \hat{R}_{tar}^{glob} & \hat{t}_{tar}^{glob} \\ 0_{1 \times 3} & 1 \end{bmatrix}^{\top}, \ T_{glob \leftarrow ego} = \begin{bmatrix} \hat{R}_{ego}^{glob} & \hat{t}_{ego}^{glob} \\ 0_{1 \times 3} & 1 \end{bmatrix} \tag{12}$$

We then obtain the transformation matrix from the ego vehicle coordinate system to the target vehicle coordinate system as:

$$T_{tar \leftarrow ego} = T_{tar \leftarrow glob} \cdot T_{glob \leftarrow ego} \tag{13}$$

To obtain the camera pose matrix in the ego vehicle coordinate system, we construct it as follows:

$$\Phi_{cam} = \begin{bmatrix} \hat{R}_{cam}^{ego} \cdot R_D & \hat{t}_{cam}^{ego} \\ 0_{1 \times 3} & 1 \end{bmatrix}, \quad R_D = \begin{bmatrix} 0 & 0 & 1 \\ 1 & 0 & 0 \\ 0 & 1 & 0 \end{bmatrix} \tag{14}$$

where $R_D$ is a transformation matrix that converts Carla's camera convention (x-forward, z-up) to PyTorch3D's camera convention (z-forward, y-up). Eventually, we compute the camera pose $\Phi_r$ in the target vehicle coordinate system, which is required as input by PyTorch3D, via:

$$\Phi_r = T_{tar \leftarrow ego} \cdot \Phi_{cam} \tag{15}$$

For multi-view rendering, We apply the above camera parameter transformation to all six ego-mounted cameras and perform image rendering as defined in Eq 1.

## A.2  PHYSICAL WORLD EXPERIMENTS

We have conducted the physical experiment to validate our method. Figure 5 shows the setup for our physical experiments. We print our generated BEVCA camouflage in a 2D UV map and stick it to a real-life-size Audi vehicle. We also built a simple multi-view camera data collection platform to collect real-life multi-view data. We set up six camera sensors on this platform, following the camera settings (positions and angles) from nuScenes. With this platform, we collect two datasets: one of the car with normal painting and another of the car with our camouflage. Each dataset has 191 samples with different azimuths and distances towards the ego position. We split both datasets based on the target vehicle distance to the ego position: 96 samples for close distance and 95 for long distance. The close distance refers to the attack vehicle being less than 10 meters from the ego position, while the long distance indicates it is more than 10 meters away.

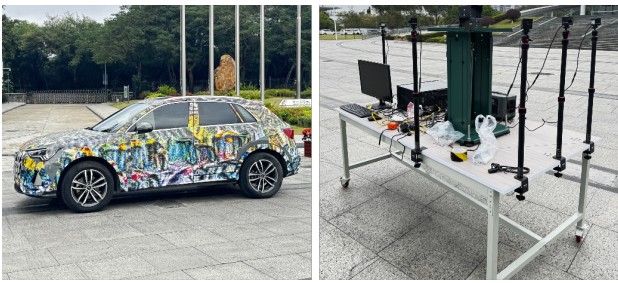

(a) Full size camouflaged car      (b) Multi-view camera platform

Figure 5: The physical experiment setup: (a) full size camouflaged car; (b) multi-view camera data collection platform.

Table 7: Comparison of the effectiveness of attacks across various original 3D detection and BEV segmentation models. The metrics for 3D detection and BEV segmentation are AP and IoU of the vehicle class, respectively.

| METHODS | 3D DETECTION | | | | | BEV SEGMENTATION | | |
|---|---|---|---|---|---|---|---|---|
| | BASE | SMALL | TINY | BASE-SEG-DET | SMALL-SEG-DET | LSS | CVT | SINBEVT |
| NORMAL | 0.309 | 0.324 | 0.243 | 0.321 | 0.333 | 0.209 | 0.234 | 0.238 |
| FCA | 0.306 | 0.336 | 0.263 | 0.32 | 0.337 | 0.204 | 0.200 | 0.208 |
| DTA | 0.260 | 0.296 | 0.249 | 0.294 | 0.318 | 0.175 | 0.168 | 0.195 |
| ACTIVE | 0.236 | 0.264 | 0.220 | 0.265 | 0.292 | 0.178 | 0.183 | 0.195 |
| RAUCA | 0.259 | 0.29 | 0.204 | 0.295 | 0.295 | 0.162 | 0.157 | 0.164 |
| CNCA | 0.227 | 0.253 | 0.185 | 0.245 | 0.284 | 0.157 | 0.201 | 0.181 |
| ADVPATCH | 0.307 | 0.310 | 0.235 | 0.321 | 0.331 | 0.193 | 0.213 | 0.202 |
| **BEVCA** | **0.080** | **0.126** | **0.158** | **0.148** | **0.171** | **0.123** | **0.065** | **0.106** |

### A.3 MULTI-VIEW ATTACK VISUALIZATION

As shown in Figure 6, we provide some examples of our camouflage attack against 3D object detection. Compared with the ground truth label, we can see that the camouflaged vehicle leads to inaccurate bounding boxes or complete "disappearance" from the detector. Our camouflage can achieve successful attacks when the target vehicle appears in not only one but also across views. As shown in Figure 7, we provide more examples of our camouflage attack against 3D object detection in the real world evaluation.

### A.4 ATTACK PERFORMANCE ON ORIGINAL TARGETED MODELS

Table 7 shows the comparison of the effectiveness of different attacks across different 3D detection and BEV segmentation original models, which are trained on the nuScenes dataset Caesar et al. (2020). Compared to Table 1, we can notice the performance gap for each attack-model pair between these two tables due to the domain gap between simulation and reality. However, BEVCA still outperforms all the baselines, proving the effectiveness and transferability of our method.

### A.5 USE OF LLMS

We only use LLMs to check grammar errors and polish writing on our draft version of the paper for the purpose of delivering our ideas and concept clearly. No use of LLMs for original work, such as idea generation.

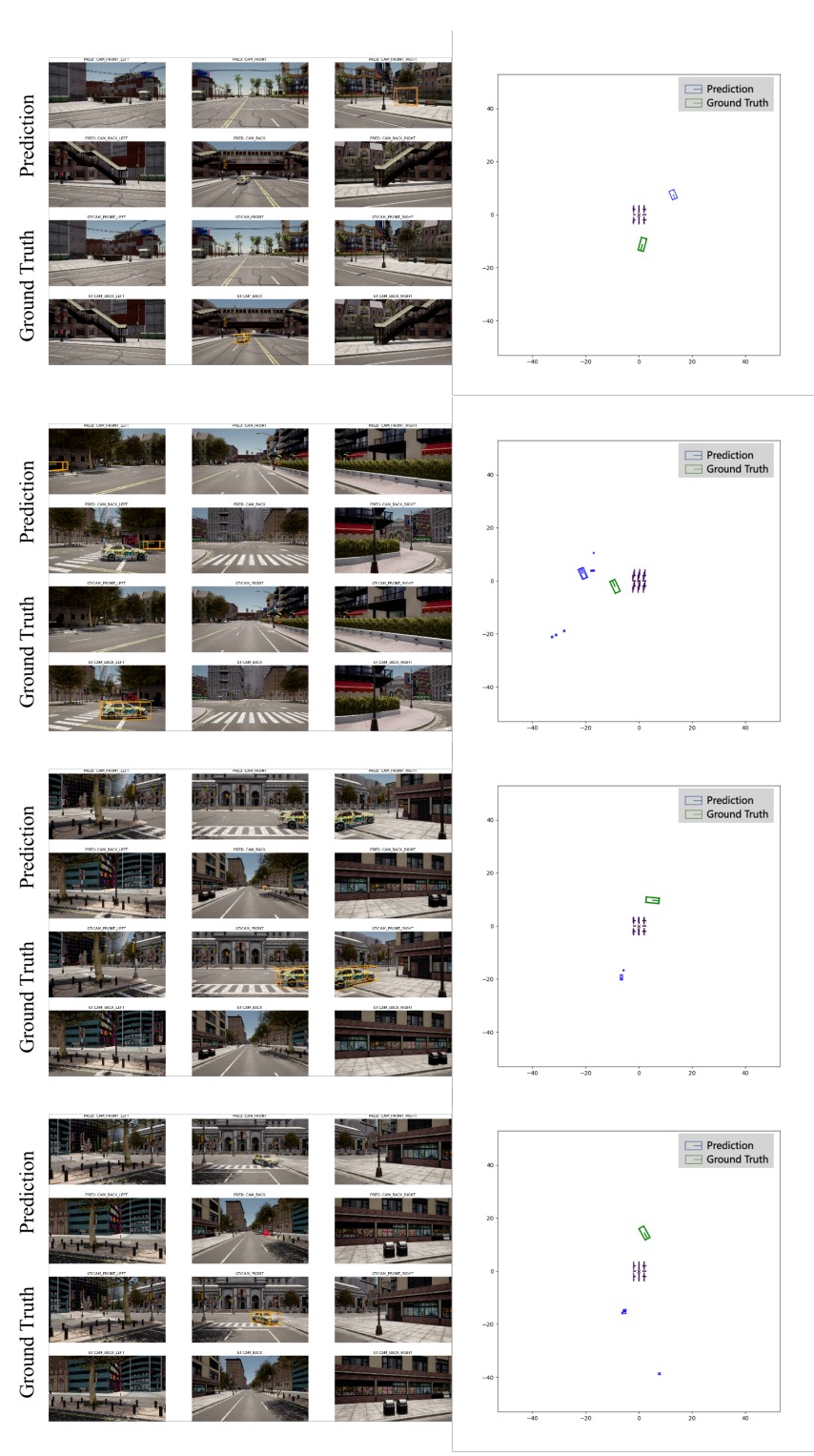

Figure 6: Visualization results of our camouflage attack against 3D object detection tasks. We can observe that the camouflaged vehicle leads to inaccurate bounding boxes or complete "disappearance" from the detector compared to the ground truths.

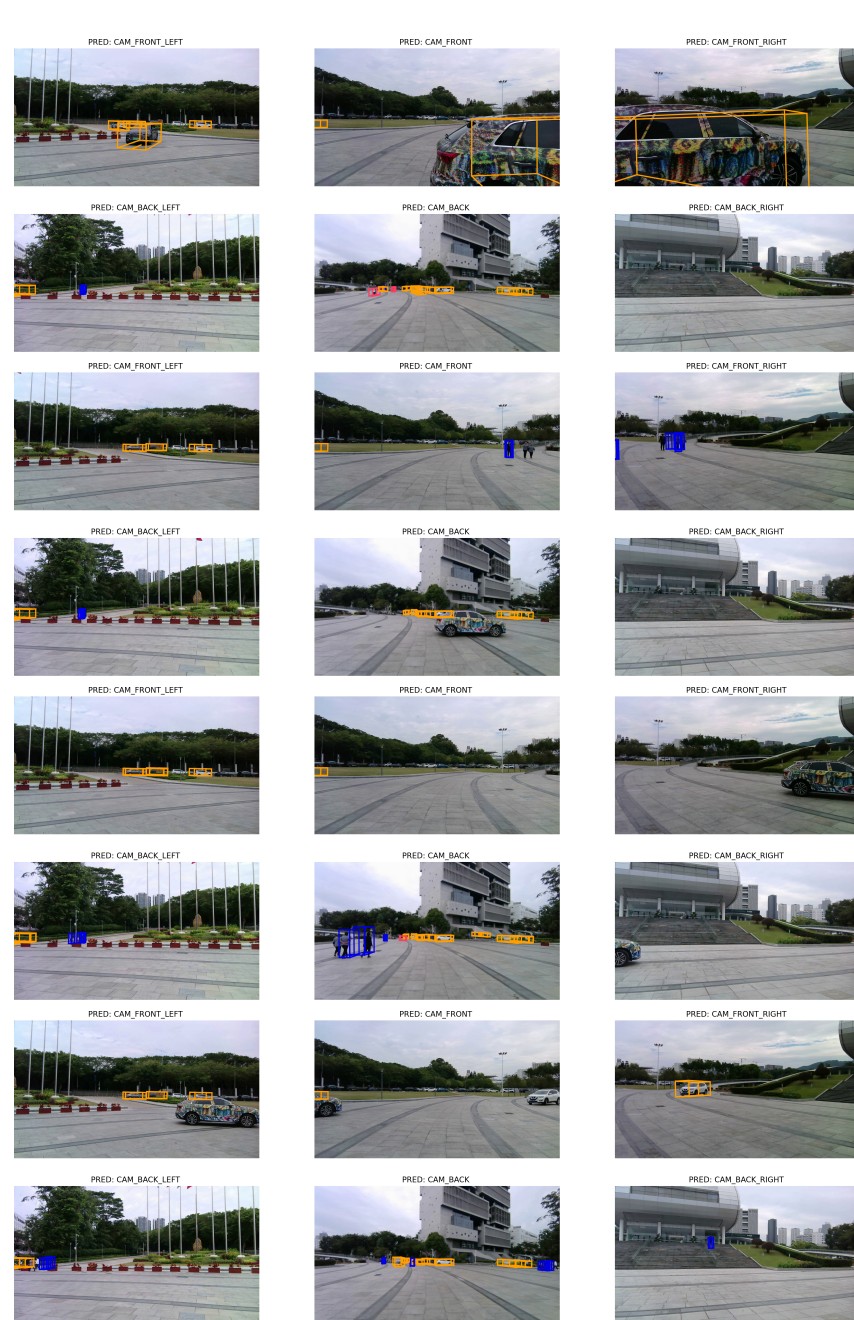

Figure 7: Additional visualization results of our camouflage attack against 3D object detection tasks in the physical world.

