# OpenReview forum: "BEVCA: Effective and Transferable Camouflage Attack against Multi-View 3D Perception in Autonomous Driving"
_ICLR.cc/2026/Conference — ICLR 2026 Conference Withdrawn Submission_

### Official Review · Reviewer_anFR · 2025-10-15

**Soundness:** 3
**Presentation:** 3
**Contribution:** 3
**Rating:** 4
**Confidence:** 4

**Summary:**

Unlike prior works that optimize adversarial textures for 2D detectors or localized patches, BEVCA introduces a multi-view neural renderer to ensure geometrically consistent textures across multiple cameras and a BEV-feature-based adversarial loss to improve transferability. Experiments on 3D object detection and segmentation tasks show significant improvements over prior baselines in both white-box and black-box settings.

**Strengths:**

1. Attacking multi-view BEV-based perception is important, since these models dominate current autonomous driving pipelines.
2. The paper evaluates across both detection and segmentation tasks, under white-box and black-box settings, with ablation studies on losses and hyperparameters.

**Weaknesses:**

1. Shifting the attack target from final outputs to early/intermediate feature spaces to improve transferability has been widely validated across vision tasks (e.g., [a-c]). Therefore, presenting “attacking BEV features” as a primary contribution is insufficient.

[a] Wang, Pengfei, et al. "Left-right Discrepancy for Adversarial Attack on Stereo Networks." arXiv preprint arXiv:2401.07188 (2024).
[b] Inkawhich, Nathan, et al. "Feature space perturbations yield more transferable adversarial examples." CVPR. 2019.
[c] Huang, Qian, et al. "Enhancing adversarial example transferability with an intermediate level attack." ICCV. 2019.

2. Whether the optimized textures are transferable to non-BEV-based 3D perception models, even 2D models, has not yet been discussed.

3. Lack of physical-world validation. All experiments are conducted in CARLA simulation, making the practicality and feasibility of BEVCA in the real world uncertain, especially given the additional noise and disturbances from real environments and sensors.

4. This paper should more thoroughly discuss the real-world feasibility of covering an entire vehicle with adversarial texture:
   * Low stealthiness. Large, conspicuous repainting or full-vehicle wraps are not as stealthy as a small patch. In practice, the attacker would likely need exclusive control of the camouflaged vehicle (e.g., their own car) to deploy the attack — it is not something an attacker can covertly apply to someone else’s vehicle. If the camouflaged vehicle is the attacker’s own car, any reduction in other vehicles’ perception increases collision risk to the attacker as well.
   * Large changes to vehicle appearance (paint, wraps, decals) are restricted or regulated in many jurisdictions and may require permits or registration changes. This creates legal barriers for real-world deployment.

**Questions:**

None

---

> ### Author Response · Authors · 2025-11-22
>
> **W1: Shifting the attack target from final outputs to early/intermediate feature spaces to improve transferability has been widely validated across vision tasks. Presenting “attacking BEV features” as a primary contribution is insufficient.**
>
> We acknowledge that the general idea of targeting feature-level representations is not new. However, applying this idea in the context of the multi-view 3D perception setting remains unexplored and challenging in current research. The first challenge is to choose which feature-level representation to attack. We particularly chose the BEV feature because many current SOTA camera-based 3D perception models utilize the BEV feature for multiple downstream tasks, even E2E autonomous driving. Therefore, targeting the BEV feature will result in a highly effective and transferable attack. The second challenge is to design an effective adversarial loss based on the BEV feature to optimize adversarial texture. Since the BEV feature is a high-dimensional representation of the environment, it is challenging to directly locate the area related to the target vehicle within the BEV feature. We solve this by introducing the mask mechanism based on the vehicle’s Ground truth location. The combination of these two technical contributions enables us to achieve an effective and transferable attack in multi-view 3D perception.
>
> **W2: Transferability to non-BEV-based 3D perception models, even 2D models, has not yet been discussed.**
>
> Our work cannot be directly transferred to non-BEV-based 3D perception and 2D models, as the scope of our work is specifically focused on attacking camera-based BEV-based 3D perception models. We choose BEV models because they are widely adopted in current end-to-end autonomous driving systems due to their high-performing effectiveness in multi-view spatial understanding. However, our framework can be adapted for non-BEV-based 3D perception models by utilizing the multi-view neural rendering module, but it requires a new adversarial loss compatible with these non-BEV-based models. Regarding 2D models, there are already many existing solutions with high performance; therefore, we did not discuss them in our paper.
>
> **W3: Lack of physical-world validation.**
>
> We have conducted the physical experiment to validate our method. We print our generated BEVCA camouflage in a 2D UV map and stick it to a real-life-size Audi vehicle. We also built a simple multi-view camera data collection platform to collect real-life multiview data. We set up six camera sensors on this platform, following the camera settings (positions and angles) from nuScenes, as most 3D perception models are trained and evaluated on the nuScenes dataset. With this platform, we collect two datasets: one of the car with normal painting and another of the car with our camouflage. Each dataset has 191 samples with different azimuths and distances towards the ego position. We split both datasets based on the target vehicle distance to the ego position: 96 samples for close distance and 95 for long distance. The close distance refers to the target vehicle being less than 10 meters from the ego position, while the long distance indicates it is more than 10 meters away. We use these two datasets to evaluate the attack performance with the Attack Success Rate (ASR) score, which measures the ratio of cases that are detected before the attack and undetected after the attack. The results are shown below:
>
> | Distance | Close | Long |
> | --- | --- | --- |
> | ASR | 53.1 % (51/96) | 15.8% (15/95) |
>
> The table shows that our BEVCA method achieves a relatively high ASR of 53.1% in close distance, demonstrating a significant threat to real-world scenarios. We have added all the relevant details of our physical experiment in our latest paper version.
>
> **W4: Real-world feasibility of covering an entire vehicle with adversarial texture.**
>
> We acknowledge that deploying large, conspicuous adversarial textures would require exclusive control over the targeted vehicle or may violate laws and regulations, making the attack less feasible. While this full camouflage attack reduces stealthiness compared to small-patch attacks, it is a highly effective and robust adversarial method that presents a realistic threat, particularly in disrupting autonomous vehicles’ perception and traffic safety.
> However, the primary motivation underlying our work is to enhance the safety and robustness of autonomous driving systems by proactively investigating potential attacks. Through a better understanding and exposure to such threats, we aim to enhance the robustness of AD systems via adversarial training, adversarial testing, and adversarial detection, thereby safeguarding the security of AD systems.

---

> > ### Comment · Reviewer_anFR · 2025-11-22
> >
> > Thank you for the responses. However, my concerns remain largely unaddressed.
> >
> > **First, regarding the contribution and challenges of attacking BEV features:**
> >
> > * As the authors acknowledge, most modern 3D perception models are BEV-based. Therefore, attacking BEV features to improve transferability is an intuitive and straightforward choice; it does not present inherent challenges. Moreover, the paper does not compare the effectiveness of attacking BEV features versus attacking other intermediate representations.
> >
> > * I agree that locating feature regions corresponding to a specific vehicle within the BEV feature map involves some complexity. However, this alone is insufficient to justify “attacking BEV features” as a major contribution of the paper.
> >
> > **Second, if the proposed attack cannot transfer to non-BEV-based multi-view 3D perception models, then the paper should avoid overstating its contribution.**
> >
> > * The paper should explicitly clarify that the proposed method applies only to BEV-based models, rather than broadly using the term “multi-view 3D perception,” since many multi-view 3D perception models are not BEV-based. This issue appears in the paper’s title, abstract, and introduction, and should be corrected to avoid misleading or overgeneral claims.
> >
> > **Lastly, while the new physical experiments demonstrate the robustness of the proposed method, fully covering a vehicle with adversarial textures remains feasible only in controlled experimental settings. Such a setup is nearly impossible to realize in real-world autonomous driving (AD) scenarios (as noted earlier in Weakness #4)**
> >
> > * The authors state that the purpose of designing this attack is to uncover potential vulnerabilities and thereby improve the safety of AD. However, the paper does not show how this attack can be concretely leveraged to enhance safety.
> >
> > * In addition, simple defenses may already be sufficient to counter this attack. For example, large-area unnatural textures can be easily detected by adversarial patch detectors (e.g., NAPGuard [d] and NutNet [e]). As a result, the real-world threat of the proposed attack remains limited, offering very limited insight for advancing AD safety.
> >
> > [d] Wu S, Wang J, Zhao J, et al. NAPGuard: Towards detecting naturalistic adversarial patches. CVPR. 2024
> >
> > [e] Lin Z, Zhao Y, Chen K, et al. I don't know you, but I can catch you: Real-time defense against diverse adversarial patches for object detectors. CCS. 2024

---

> ### Author Response · Authors · 2025-12-04
>
> **C1. Contribution and challenges of attacking BEV features.**
>
> Thank you for your thoughtful comments. While attacking BEV features may appear intuitive because modern 3D perception models share a BEV representation, **achieving an effective BEV-level attack is far from straightforward**. In practice, a naïve BEV-feature attack performs poorly due to substantial noise from car-irrelevant regions in the BEV map. As you suggested, we empirically compare BEV-feature attacks with two alternative intermediate representations—image features (L_feats) and detection-head hidden states (L_det_hs). As shown in the table below, **attacking BEV features only becomes effective when we solve the key domain-specific challenge of identifying car-texture-sensitive BEV regions**, which we address via our proposed masking technique.
>
> | **Loss Type** | **BEVFormer-base** | **BEVFormer-small** | **BEVFormer-tiny** | **BEVFormer-base-seg-det** | **BEVFormer-small-seg-det** | **LSS** | **CVT** | **SinBEVT** |
> | --- | --- | --- | --- | --- | --- | --- | --- | --- |
> | L_feats | 0.392 | 0.564 | 0.428 | 0.528 | 0.470 | 0.380 | 0.371 | 0.333 |
> | L_det_hs | 0.471 | 0.603 | 0.485 | 0.601 | 0.529 | 0.367 | 0.426 | 0.347 |
> | L_bev + no mask | 0.612 | 0.704 | 0.508 | 0.735 | 0.585 | 0.498 | 0.588 | 0.574 |
> | **L_bev + mask** | **0.191** | **0.285** | **0.381** | **0.329** | **0.329** | **0.253** | **0.307** | **0.258** |
>
> These results show the following key findings:
>
> - **Attacking BEV features without addressing car-relevant region localization fails** (L_bev + no mask).
> - **Other intermediate representations are significantly less effective than BEV features.**
> - **Only our masked BEV-feature loss produces strong and transferable attacks across diverse BEV-based models.**
>
> Thus, the challenge is not choosing BEV features, but **making BEV-level optimization effective**, which requires solving a domain-specific problem—**locating the precise BEV feature regions influenced by vehicle texture**. Our heuristic masking method, derived from analyzing BEV differences in with-car vs. no-car scenes, is essential and novel in enabling BEV-based texture attacks.
>
> Beyond BEV-feature optimization, our work introduces **additional domain-specific contributions** that are not addressed by prior 3D texture-based attacks:
>
> 1. **A multi-view neural rendering module** that enables accurate, consistent rendering of textured vehicles across multiple camera views. Prior 3D texture methods operate in single-view settings; multi-view rendering requires handling six additional camera transformations and ensuring geometric consistency—critical for gradient-based texture optimization in multi-view perception models.
>
> 2. **The first full-size, multi-view physical vehicle camouflage attack.** Previous works evaluate on small toy cars, whereas we construct a full multi-view data collection system, print and apply adversarial camouflage to a real vehicle, and perform road testing. This engineering effort is non-trivial and demonstrates real-world feasibility at full scale.
>
>
> In summary, although the high-level idea of attacking BEV features may seem intuitive, **effective BEV-feature attacks require solving challenges that have not been addressed in prior work**, including (1) identifying car-relevant BEV representations, (2) enabling multi-view differentiable rendering, and (3) validating on full-size real vehicles. These contributions collectively advance adversarial research in multi-view BEV-based perception beyond prior 3D texture approaches.
>
> **C2: Avoid overstating contribution.**
>
> We thank the reviewer for this valuable comment. We agree that our proposed attack framework specifically targets BEV-based multi-view 3D perception. Following the reviewer’s suggestion, we have revised the paper’s title, abstract, and introduction to avoid potential overgeneralization. Please refer to our latest paper version for this update.

---

> > ### Author Response · Authors · 2025-12-04
> >
> > **C3.1. Leverage our attack to enhance safety.**
> >
> > We appreciate the suggestion to clarify the concrete safety implications of our work. Beyond identifying vulnerabilities, our attack can serve as a practical method for **enhancing the robustness of BEV-based perception models via adversarial training**. To demonstrate this, we generate an adversarial dataset of 500 samples with our camouflage. We use different physical configurations to generate the adversarial dataset, ensuring it differs from the test set. We then fine-tune eight state-of-the-art BEV perception models using this dataset. The results, summarized in the table below:
> >
> > | **Metric / Model** | **BEVFormer-base** | **BEVFormer-small** | **BEVFormer-tiny** | **BEVFormer-base-seg-det** | **BEVFormer-small-seg-det** | **LSS** | **CVT** | **SinBEVT** |
> > | --- | --- | --- | --- | --- | --- | --- | --- | --- |
> > | Normal Performance | 0.618 | 0.721 | 0.514 | 0.750 | 0.634 | 0.509 | 0.609 | 0.570 |
> > | Under Attack | 0.191 | 0.285 | 0.381 | 0.329 | 0.331 | 0.253 | 0.307 | 0.273 |
> > | After Adversarial Training | 0.615 | 0.479 | 0.416 | 0.683 | 0.599 | 0.614 | 0.567 | 0.549 |
> >
> > **Key Findings of the table results are the following:**
> >
> > - As shown in rows of ‘Normal Performance’ and ‘Under Attack’, **the original models experience a significant drop in performance when exposed to our adversarial camouflage.** For example, BEVFormer-base's detection score falls from 0.618 to 0.191 and CVT drops from 0.609 to 0.307,  highlighting the vulnerability of current perception systems to physical camouflage attacks.
> > - As shown in the rows of 'Under Attack' and 'After Adversarial Training', **after adversarial training, all models recover a substantial portion of their detection capability.** For instance,  BEVFormer-base’s detection score improves from 0.191 to 0.615 and CVT’s score increases from 0.307 to 0.567.
> >
> > These results demonstrate that our attack can be directly used to significantly improve model robustness.
> >
> > **C3.2. Counter our attack with adversarial texture detection.**
> >
> > Thanks for your comment. As requested, we conducted experiments using the official implementations of **NAPGuard** and **NutNet** against our attack. We evaluated these defenses using our physical experiment datasets: the **Normal-Paint Car** dataset (clean) and the **BEVCA Camouflaged Car** dataset (adversarial). Both datasets comprise 191 samples.
> >
> > Our results indicate that **neither defense is effective in countering our 3D camouflage attack**. Specifically, both methods **fail to accurately detect the adversarial texture due to a high false negative rate.**
> >
> > **1. Analysis of NAPGuard**
> >
> > NAPGuard is designed to detect adversarial patches by identifying aggressive feature modulations. We applied NAPGuard to both our normal and adversarial datasets to evaluate its detection capability.
> >
> > | **Dataset** | **Ground Truth (Has Patch?)** | **NAPGuard Detected Rate** | **Result** |
> > | --- | --- | --- | --- |
> > | **Normal Dataset** | No | **0%** | Low False Positive |
> > | **Adversarial Dataset** | Yes | **4%** | **High False Negative** |
> >
> > As shown in the table, while NAPGuard correctly identifies normal images (0% false positive rate), it detects the adversarial camouflage in only **4%** of the adversarial images. This suggests that our camouflage method does not exhibit the specific "aggressive" 2D feature that NAPGuard is trained to detect.
> >
> > **2. Analysis of NutNet**
> >
> > **NutNet** is an autoencoder trained to reconstruct images sampled from the normal distribution. We evaluate its detection capability by calculating reconstruction errors (MSE) between the input and output images. A smaller reconstruction error indicates that the input image is more likely to belong to the normal distribution, while a larger error suggests the image deviates from normality and potentially contains adversarial perturbations. We evaluated NutNet's performance by computing reconstruction errors for both normal and adversarial datasets. When we set the detection threshold to the maximum reconstruction error observed in the normal dataset, aiming to achieve zero false positives (i.e., filtering out all normal samples), only 58% of adversarial examples are successfully detected. Critically, **42% of adversarial examples remain undetected**, as their reconstruction errors fall within or below the normal range. This high false negative rate indicates that **NutNet is unable to effectively detect our camouflaged adversarial attacks.**

---

> ### Author Response · Authors · 2025-12-04
> **Summary of Rebuttal: Reviewer anFR**
>
> Reviewer anFR acknowledged **the importance of our work in attacking multi-view BEV-based perception and conducting comprehensive evaluations on different tasks under both white-box and black-box settings**. However, the reviewer raised concerns about the challenges and contributions of the BEVCA attack, as well as the practicality of using our attack in the real world. Specifically, they questioned how to utilize our attack to enhance safety and whether simple defenses, such as adversarial texture detection, could counter our attack.
>
> > ### **Concern 1: Justify BEVCA attack challenges and contributions**
>
> - **Solution:**  We provide additional ablation studies by comparing the BEV-feature attack with two alternative intermediate representations: image features and detection-head hidden states. The results show that **attacking BEV features only becomes effective when we solve the key domain-specific challenge of identifying car-texture-sensitive BEV regions**, which we address via our proposed masking technique. Thus, the challenge is not choosing BEV features, but **making BEV-level optimization effective**, which requires solving a domain-specific problem—**locating the precise BEV feature regions influenced by vehicle texture**. Besides, our work also addresses **additional domain-specific contributions** that are not addressed by prior 3D texture-based attacks: (1) **A multi-view neural rendering module** that enables accurate, consistent rendering of textured vehicles across multiple camera view; (2) **The first full-size, multi-view physical vehicle camouflage attack.**
> - **Conclusion:** In summary, although the high-level idea of attacking BEV features may seem intuitive, **effective BEV-feature attacks require solving challenges that have not been addressed in prior work**, including (1) **identifying car-relevant BEV representations, (2) enabling multi-view differentiable rendering, and (3) validating on full-size real vehicles.** These contributions collectively advance adversarial research in multi-view BEV-based perception beyond prior 3D texture approaches.
>
> > ### **Concern 2: How to use our attack to enhance safety**
>
> - **Solution:** We conduct additional experiments of **enhancing the robustness of BEV-based perception models via adversarial training.** We generate a dedicated adversarial training to enhance the original BEV-based perception models. The results of the experiments show that **the original models experience a significant drop in performance when exposed to our adversarial camouflage. After adversarial training, all models recover a substantial portion of their detection capability**
> - **Conclusion:** The adversarial training experiments demonstrate that our attack can be directly used to significantly improve model robustness.
>
> > ### **Concern 3: Simple defense with adversarial texture detection**
>
> - **Solution:** As suggested, we conducted experiments using the official implementations of **NAPGuard** and **NutNet** against our attack. We evaluated these defenses using our physical experiment datasets: the **Normal-Paint Car** dataset (clean) and the **BEVCA Camouflaged Car** dataset (adversarial). Our results indicate that **neither defense is effective in countering our 3D camouflage attack**. Specifically, both methods fail to accurately detect the adversarial texture due to a high false-negative rate.**
> - **Conclusion:** Our experiments with adversarial texture detectors demonstrate that our proposed attack cannot be effectively countered by simple defense methods like adversarial texture detection.

---

### Official Review · Reviewer_4tc6 · 2025-10-30

**Soundness:** 3
**Presentation:** 3
**Contribution:** 3
**Rating:** 6
**Confidence:** 3

**Summary:**

I like the proposed ideas in this paper, e.g., BEV masking and incentivizing the features to get to zero. I also like that it seems from the results to be transferrable. however I would like further expansion as to why this attack is transferrable unlike other attacks.

The results are solid; however experiments confined to Carla are not ideal considering the many prior works that worked directly on realistic scenes. This puts into question the sim2real gap and the applicability of this work in the real world.

**Strengths:**

The strengths of this paper lie in:
1. I like the idea that the adversarial loss is on the BEV feature instead of the detection output. Focusing on zeroing out that BEV feature is pretty cool and makes sense; though there should've been a lot more expansion about why that might be a good idea and how the authors came up with it and how it helps in black-box settings.
2. I like the novelty of the black box attack which isn’t very common and a much harder problem.
3. The focus on camouflage instead of patching is good, but the authors should mention other approaches as well like 3D objects [1] which can rendered a lot easier in real scenes instead of needed models of Cars to fit the camouflage on.
4. The results are pretty good and attractive, though the dataset is synthetic.

**Weaknesses:**

The paper has some nice results, but in my opinion doesn’t have wide applicability for a few reasons:
1. Everything is simulated in Carla no real life datasets (the work [1] from 4 years ago was also 3D rendered but on the real KITTI dataset) this could put into question the applicability of this work to real life threats.
2. The dataset is synthetic and custom made and baselines are evaluated against this custom 500 scene dataset which can make the result hard to interpret considering there are many existing baseline out there.
3. The qualitative examples seem to mostly show the attacking car in isolation and no other cars or objects around. Also the cars are placed awkwardly and randomly in the scene. This is extremely unrealistic and again puts into question if this method could pose a real threat relative to the numbers.
4. Level of novelty: camouflage attacks have been around but the novelty is mainly in the BEV adversarial loss (expanded on in strengths).

[1] Abdelfattah, Mazen, et al. "Adversarial attacks on camera-lidar models for 3d car detection." 2021 IEEE/RSJ International Conference on Intelligent Robots and Systems (IROS). IEEE, 2021.

**Questions:**

What would’ve made this result very strong is the rendering of this car on real multi-view NuScenes scenes and reporting how SOTA models struggled.

---

> ### Author Response · Authors · 2025-11-22
>
> **W1: Only studied in Carla simulation with no real-life datasets.**
>
> Thanks for your valuable feedback. First, the work [1] is a digital attack based on a single-view image, which cannot be applied in the real world on the vehicle’s surface. In our work, we are actually using adversarial camouflage to fully cover the car to achieve the attack in the real world.
>
> We have conducted the physical experiment to validate our method. We print our generated BEVCA camouflage in a 2D UV map and stick it to a real-life-size Audi vehicle. We also built a simple multi-view camera data collection platform to collect real-life multiview data. We set up six camera sensors on this platform, following the camera settings (positions and angles) from nuScenes, as most 3D perception models are trained and evaluated on the nuScenes dataset. With this platform, we collect two datasets: one of the car with normal painting and another of the car with our camouflage. Each dataset has 191 samples with different azimuths and distances towards the ego position. We split both datasets based on the target vehicle distance to the ego position: 96 samples for close distance and 95 for long distance. The close distance refers to the target vehicle being less than 10 meters from the ego position, while the long distance indicates it is more than 10 meters away. We use these two datasets to evaluate the attack performance with the Attack Success Rate (ASR) score, which measures the ratio of cases that are detected before the attack and undetected after the attack. The results are shown below:
>
> | Distance | Close | Long |
> | --- | --- | --- |
> | ASR | 53.1 % (51/96) | 15.8% (15/95) |
>
> The table shows that our BEVCA method achieves a relatively high ASR of 53.1% in close distance, demonstrating a significant threat to real-world scenarios. We have added all the relevant details of our physical experiment in our latest paper version.
>
> [1] Abdelfattah, Mazen, et al. "Adversarial attacks on camera-lidar models for 3d car detection." 2021 IEEE/RSJ International Conference on Intelligent Robots and Systems (IROS). IEEE, 2021.
>
> **W2: The test dataset is synthetic instead of using the existing baselines.**
>
> Directly evaluating our method on existing real-world baselines, such as nuScenes, presents substantial technical challenges. Our approach relies on generating adversarial camouflage as a 3D texture, which requires rendering onto a compatible 3D car model to produce the 2D images for detector input. However, the nuScenes dataset, while containing various vehicle scenarios, does not provide the associated 3D models needed for this process. Therefore, it is infeasible to render our camouflage in a manner that faithfully aligns with the nuScenes dataset images. Overcoming this challenge would likely require either augmenting datasets with explicit 3D geometry for each vehicle or developing new methods to approximate such correspondence, both of which remain open problems at this moment.
>
> The current research communities for physical adversarial camouflage [1][2][3][4][5] follow the similar procedures to assess the generated camouflage's attack performance: first, apply the adversarial camouflage onto the car in both simulated and realistic environments; then sample them with various physical conditions such as distances, angles, lightings, etc to build the test datasets; lastly, the test datasets are evaluated by the target models to validate the effectiveness and transferability of the attacks.
>
> [1]  Wang, Donghua, et al. "Fca: Learning a 3d full-coverage vehicle camouflage for multi-view physical adversarial attack." *Proceedings of the AAAI conference on artificial intelligence*. Vol. 36. No. 2. 2022.
>
> [2] Suryanto, Naufal, et al. "Dta: Physical camouflage attacks using differentiable transformation network." *Proceedings of the IEEE/CVF Conference on Computer Vision and Pattern Recognition*. 2022.
>
> [3] Suryanto, Naufal, et al. "Active: Towards highly transferable 3d physical camouflage for universal and robust vehicle evasion." *Proceedings of the IEEE/CVF international conference on computer vision*. 2023.
>
> [4] Zhou, Jiawei, et al. "Rauca: A novel physical adversarial attack on vehicle detectors via robust and accurate camouflage generation." *arXiv preprint arXiv:2402.15853* (2024).
>
> [5] Lyu, Linye, et al. "CNCA: Toward Customizable and Natural Generation of Adversarial Camouflage for Vehicle Detectors." *arXiv preprint arXiv:2409.17963* (2024).

---

> ### Author Response · Authors · 2025-11-22
>
> **W3: The qualitative examples show the attacking car in isolation and no other cars or objects around, which is unrealistic.**
>
> We acknowledge that the single-car scenarios in our current simulation dataset are not so realistic. However, as this is the first work to generate adversarial camouflage targeted at camera-based 3D perception models, we began with simplified settings featuring a single target vehicle to maintain better experimental control. Our newly added physical-world experiment has addressed this concern. As illustrated in Figure 4 in our latest paper version, our test dataset captures various objects besides the target vehicle, such as other cars, scooters, and pedestrians. The results show that, with our attack, the target car is not detected, while other objects remain reliably detected by the detector. This outcome highlights the real-world threat posed by our method. We greatly appreciate this feedback and plan to incorporate more diverse objects into our future work to further enhance the realism of our scenarios.
>
> **W4: Level of novelty: camouflage attacks have been around, but the novelty is mainly in the BEV adversarial loss.**
>
> We would like to stress that our work is the first framework to attack multi-view camera-based 3D perception models with full-cover adversarial camouflage. To achieve this, we adapt the current 2D attack pipeline to 3D detection tasks with two novel contributions. Specifically, we first introduce a new multi-view neural rendering module to enable gradient optimization towards the 3D adversarial texture. Secondly, we design a novel BEV-based adversarial loss with a mask mechanism to accurately locate the target vehicle BEV region. The combination of these novel contributions enables us to achieve an effective and transferable attack.
>
> **Q1: Rendering of this car on real, multi-view NuScenes scenes and reporting how SOTA models struggled.**
>
> As discussed in W1, it is not feasible to directly render our camouflage onto the nuScenes dataset because nuScenes lacks the 3D car models required to synthesize 2D vehicle images for attacking the detector. To properly evaluate our method, we first apply the generated camouflage to compatible car models, then capture images of the camouflaged vehicles under varying conditions—such as different distances, angles, and lighting—in both simulation and real-world environments. After obtaining these evaluation datasets, we can use them to assess the target model’s detection performance by comparing its results before and after the attack. By reporting the observed degradation in detection performance, we can demonstrate the effectiveness of our proposed attack.

---

> ### Author Response · Authors · 2025-12-04
> **Summary of Rebuttal: Reviewer 4tc6**
>
> Reviewer 4tc6 acknowledged **the novelty of the BEV-feature-based adversarial loss design and good attack performance in black-box settings** of our proposed BEVCA method. However, the reviewer raised concerns about the effectiveness of our attack in real-world situations.
>
> > ### **Concern 1: Require real-world physical world validation**
>
> - **Solution:** As required, we present the detailed physical world experiments with our camouflage methods, including both experiment setup and results (**see Table 3 and Figure 4 in the latest paper version**). The physical world experiments validate the robustness of our attack in the real world.
> - **Conclusion:** The physical world experiments have validated the effectiveness of our proposed BEVCA on real-world scenarios.

---

### Official Review · Reviewer_BwNt · 2025-11-01

**Soundness:** 2
**Presentation:** 3
**Contribution:** 2
**Rating:** 4
**Confidence:** 4

**Summary:**

This paper introduces a new method that creates camouflage patterns to fool the multi-camera 3D perception systems used in self-driving cars. By attacking the system's internal BEV representation, the attack remains effective across different models and tasks without needing to know their specific designs.

**Strengths:**

1. Good attack performance: The proposed BEVCA method largely outperforms all existing baselines.

2 . Clear presentation: The paper's core idea and methodology are explained clearly, and the writing is easy to follow.

**Weaknesses:**

1. Experimental comparison issues: (1) The comparison results with methods like ACTIVE are not meaningful, because these methods were trained using 2D object detection loss functions. Directly applying them to attack 3D object detection is unfair. Therefore, the Multiview Neural Rendering proposed in this paper does not necessarily outperform previous methods. (2) The paper cites NeRF-based 3D adversarial texture methods (e.g., Li et al., 2024) but does not compare their method to them.

2. Physical-world validation: As noted in Section 5 (Conclusion & Limitation) of the paper: "Our current work is limited… as we have not conducted physical experiments…." However, physical-world testing is essential in this field.

3. Feature visualization: The visualization of the BEV difference features in the paper could be enhanced. It is suggested to add more intuitive indicators and a more detailed explanation.

**Questions:**

see weaknesses

---

> ### Author Response · Authors · 2025-11-22
>
> **W1: Comparison results with methods like ACTIVE are not meaningful. The NeRF-based method is not compared.**
>
> We are the first work to generate full-cover camouflage to attack 3D detection models with no directly comparable baselines. Since there are many existing camouflage methods for 2D detection, we select them as references to generate multi-view test datasets to attack 3D detection models. However, these 2D attack baselines cannot be easily adapted to attack 3D detection models. They need the following adaptation:
>
> 1. Adapt the single-view 2D vehicle image dataset to a multi-view vehicle dataset.
> 2. Adapt the single-view neural rendering module to the multi-view neural rendering module.
> 3. Instead of directly using the downstream task score as an adversarial loss, design a novel loss function to achieve effective and transferable attacks in 3D perception tasks.
>
> We address these challenges primarily by proposing a new multi-view neural rendering module and a novel BEV-feature-based adversarial loss, which enable an effective and transferable attack. These are the main contributions of our work.
>
> Regarding the NeRF-based 3D adversarial texture method Adv3D [1], we find that this approach generates adversarial examples based on single-view images rather than multi-view images. Currently, the major car manufacturers, such as Tesla and XPeng, deploy multi-view perception models for their autonomous driving systems. More importantly, this method only generates an adversarial patch for one side of the vehicle (as demonstrated in their real-world scenario), which limits the robustness of the attack in the physical world. We want to focus on full-cover adversarial camouflage attacks because they have been proven to be more multi-view robust and effective in the real world. Therefore, we do not include this method in our experiments.
>
> [1] https://len-li.github.io/adv3d-web/
>
> **W2: Physical-world validation.**
>
> We have conducted the physical experiment to validate our method. We print our generated BEVCA camouflage in a 2D UV map and stick it to a real-life-size Audi vehicle. We also built a simple multi-view camera data collection platform to collect real-life multiview data. We set up six camera sensors on this platform, following the camera settings (positions and angles) from nuScenes, as most 3D perception models are trained and evaluated on the nuScenes dataset. With this platform, we collect two datasets: one of the car with normal painting and another of the car with our camouflage. Each dataset has 191 samples with different azimuths and distances towards the ego position. We split both datasets based on the target vehicle distance to the ego position: 96 samples for close distance and 95 for long distance. The close distance refers to the target vehicle being less than 10 meters from the ego position, while the long distance indicates it is more than 10 meters away. We use these two datasets to evaluate the attack performance with the Attack Success Rate (ASR) score, which measures the ratio of cases that are detected before the attack and undetected after the attack. The results are shown below:
>
> | Distance | Close | Long |
> | --- | --- | --- |
> | ASR | 53.1 % (51/96) | 15.8% (15/95) |
>
> The table shows that our BEVCA method achieves a relatively high ASR of 53.1% in close distance, demonstrating a significant threat to real-world scenarios. We have added all the relevant details of our physical experiment in our latest paper version.
>
> **W3: Feature visualization improvement.**
>
> Thanks for your suggestion. We have enhanced the BEV difference feature visualization with improved indicators and added an explanatory section in Figure 3 of our latest paper version.

---

> > ### Comment · Reviewer_BwNt · 2025-11-25
> > **Thanks**
> >
> > The reviewer appreciates the new real-world results. However, the paper suffers from limited novelty (widely used feature-level attack) and the fundamental similarity to other 3D texture-based attacks (in other tasks). Therefore, the reviewer does not see any domain-specific challenges in applying such existing general techniques/formulations to this new task.

---

> ### Author Response · Authors · 2025-12-04
>
> Thanks for your comments. We respectfully disagree with the assessment that our work offers limited novelty. To the best of our knowledge, **our method is the first framework capable of generating adversarial camouflage against multi-view, BEV-based 3D perception models**, representing **a substantive advance in adversarial research for vehicle detection**. Unlike prior feature-level attacks or single-view 3D texture attacks, our work explicitly tackles **three core challenges inherent to attacking multi-view BEV-based 3D detection systems**:
>
> 1. **Achieving an effective and transferable attack requires solving two coupled problems:** (i) identifying an appropriate intermediate representation for adversarial optimization, and (ii) discovering the *car-texture-sensitive* feature regions within that representation to obtain meaningful gradients for texture optimization.
> 2. **Existing 3D texture-based methods support only single-view rendering**, making them unsuitable for accurate and consistent multi-view adversarial camouflage generation.
> 3. **Prior physical experiments have only used small-scale car models**, leaving full-scale adversarial camouflage for real vehicles unexplored.
>
> Our paper addresses these challenges through the following key contributions:
>
> 1. **We propose a BEV-feature-level attack enhanced with a dedicated masking strategy that yields highly effective and transferable adversarial camouflage.** Multi-view 3D perception models rely heavily on **BEV features** as the shared spatial representation for downstream tasks; thus, attacking BEV features provides strong transferability across architectures. However, using the entire BEV feature map for loss computation introduces noise from **car-texture-insensitive regions**, which suppresses meaningful gradients and weakens the attack.
>
>     To overcome this issue, we analyze **BEV feature differences** between scenes with and without a target vehicle, revealing a strong alignment between **car-texture-sensitive BEV regions and ground-truth vehicle locations**. This observation motivates our **heuristic BEV masking technique**, which accurately isolates car-texture-relevant features for adversarial optimization.
>
>     To validate the necessity of both (i) BEV-level optimization and (ii) our masking scheme, we compare against two alternative feature-selection strategies:
>
>     (1) image-encoder outputs (L_feats),
>
>     (2) hidden-state features from the 3D detection head (L_det_hs),
>
>     and also evaluate targeting BEV features *without* masking (L_bev + no mask).
>
>     | **Loss Type** | **BEVFormer-base** | **BEVFormer-small** | **BEVFormer-tiny** | **BEVFormer-base-seg-det** | **BEVFormer-small-seg-det** | **LSS** | **CVT** | **SinBEVT** |
>     | --- | --- | --- | --- | --- | --- | --- | --- | --- |
>     | L_feats | 0.392 | 0.564 | 0.428 | 0.528 | 0.470 | 0.380 | 0.371 | 0.333 |
>     | L_det_hs | 0.471 | 0.603 | 0.485 | 0.601 | 0.529 | 0.367 | 0.426 | 0.347 |
>     | L_bev + no mask | 0.612 | 0.704 | 0.508 | 0.735 | 0.585 | 0.498 | 0.588 | 0.574 |
>     | **L_bev + mask** | **0.191** | **0.285** | **0.381** | **0.329** | **0.329** | **0.253** | **0.307** | **0.258** |
>
>     These results show that **non-BEV features yield weak attack performance**, and **attacking BEV features without masking is ineffective**. In contrast, our **BEV-level attack with masking achieves the strongest and most transferable effects across diverse BEV-based perception models**, demonstrating the necessity and effectiveness of our design choices.
>
> 2.  **We propose a new multi-view neural rendering module that enables accurate and consistent rendering of textured vehicles across diverse camera viewpoints in multi-view datasets.** This module overcomes a key engineering bottleneck—reliably rendering the same vehicle under varying camera configurations. To accomplish this, we perform two coordinate transformations across three coordinate systems (Carla, nuScenes, and PyTorch3D), ensuring precise geometric alignment and consistent multi-view projection. This design **supports stable gradient-based texture optimization against multi-view perception models** and can also **serve as a general-purpose multi-view neural rendering tool** for a wide range of vision and perception research tasks.
>
> 3. **We are also the first, to our knowledge, to conduct multi-view, full-size physical vehicle camouflage attack experiments.** Executing full-scale physical attacks requires significant engineering effort, including constructing a multi-view camera data collection platform, printing and applying adversarial camouflage onto a full-size vehicle, and organizing road testing to acquire multi-view datasets. The scale and rigor of these experiments substantially exceed prior small-model demonstrations, and we believe that our experimental design, infrastructure, and execution represent a meaningful contribution to the research community.

---

> ### Author Response · Authors · 2025-12-04
> **Summary of Rebuttal: Reviewer BwNt**
>
> Reviewer UpWB acknowledged that our proposed BEVCA method achieves **good attack performance, outperforming all existing baselines**, and the **clear presentation of the paper**. However, the reviewer raised concerns about the **limited novelty and similarity to other 3D texture-based attacks.**
>
> > ### **Concern 1: limited novelty and similarity to other 3D texture attacks**
>
> - **Solution:** We stress that our proposed BEVCA method **is the first framework capable of generating adversarial camouflage against multi-view, BEV-based 3D perception models.** Unlike prior attack methods, we need to address the following challenges: **(1) Achieving an effective and transferable attack; (2) Existing 3D texture-based methods support only single-view rendering; (3) Prior physical experiments have only used small-scale car models.** Our paper addresses these challenges through the following key contributions: **(1) We propose a BEV-feature-level attack enhanced with a dedicated masking strategy that yields highly effective and transferable adversarial camouflage, supported by ablation studies of different adversarial loss designs. (2) We propose a new multi-view neural rendering module that enables accurate and consistent rendering of textured vehicles across diverse camera viewpoints in multi-view datasets. (3) We are also the first, to our knowledge, to conduct multi-view, full-size physical vehicle camouflage attack experiments.**
>
> - **Conclusion:**  Our method is the first framework capable of generating adversarial camouflage against multi-view, BEV-based 3D perception models, representing a substantive advance in adversarial research for vehicle detection.

---

### Official Review · Reviewer_UpWB · 2025-11-07

**Soundness:** 3
**Presentation:** 3
**Contribution:** 3
**Rating:** 4
**Confidence:** 3

**Summary:**

This work proposes BEVCA, the first work to generate adversarial camouflage to attack multi-view 3D BEV-based detection models. Its framework consists of two core novel modules: multi-view neural rendering and BEV-feature-based attack modules. The former module conduct various coordinate system transformation to generate ego-vehicle multi-view images and use differentiable rendering library PyTorch3D to support gradient-based attack. The latter module introduces novel attack on the BEV feature to increase effectiveness and transferrability. Experiments on Carla simulated environment with both white-box and black-box setting show non-trivial success rate and reveal severe security threat in existing multi-view 3D perception models.

**Strengths:**

Study on attacking 3D detectors with camouflage is a very practical and valuable research topic with lots of applications in autonomous driving. This work is the first to generate adversarial camouflage that effectively attacks multi-view 3D perception models. On simulated data, BEVCA shows non-trivial attack success rate, revealing clear security loophole in existing autonomous driving systems.

**Weaknesses:**

1. This paper only studies simulated environment, but never present any real-world attack successful case. For instance, the neural rendering module is specifically designed for simulation data, making it hard to bridge the sim2real gap. If this work cannot be applied to real-world data, its value and practibility is greatly compromised. I wonder if the authors have given any thoughts on how to adapt BEVCA to real data (e.g. nuScenes)?

2. Only BEVFormer is studied in this work. As another line of representative BEV-based multi-view 3D detector, BEVDepth[1] should also be included.

**References**

[1] Li, Yinhao, et al. "Bevdepth: Acquisition of reliable depth for multi-view 3d object detection." Proceedings of the AAAI conference on artificial intelligence. Vol. 37. No. 2. 2023.

**Questions:**

1. The authors use a synthetic fine-tuning dataset of 500 samples to bridge the real2sim domain gap, which is several orders magnitude less than the original nuScenes dataset. I wonder if that is enough to adapt the model to the Carla Simulator domain.

2. It is a bit counter-intuitive to see that optimizing L_det or L_bev + No Mask leads to almost no performance degradation in Table 3. In my previous experience, white-box attack almost always break a non-adversarial model after sufficient optimization steps. The authors claim that substantial noise signals leads to ineffective noise signals, which I find rather hard to believe. Well-optimized attack images should cover that even it contains a lot of background parts. I wonder if the authors could give some explanation on this result?

3. In the black-box setting, how was the transferability evaluated? I've only seen a bunch of 3D object detection and BEV segmentation
models being used (line 335), but what models are used as source model, and what is used as target model?

---

> ### Author Response · Authors · 2025-11-22
>
> **W1: only studies the simulation environment without real-world attacks. Adapt BEVCA to real data like nuScenes?**
>
> Thank you for your valuable feedback and thoughtful comments. We have conducted the physical experiment to validate our method. We print our generated BEVCA camouflage in a 2D UV map and stick it to a real-life-size Audi vehicle. We also built a simple multi-view camera data collection platform to collect real-life multiview data. We set up six camera sensors on this platform, following the camera settings (positions and angles) from nuScenes, as most 3D perception models are trained and evaluated on the nuScenes dataset. With this platform, we collect two datasets: one of the car with normal painting and another of the car with our camouflage. Each dataset has 191 samples with different azimuths and distances towards the ego position. We split both datasets based on the target vehicle distance to the ego position: 96 samples for close distance and 95 for long distance. The close distance refers to the target vehicle being less than 10 meters from the ego position, while the long distance indicates it is more than 10 meters away. We use these two datasets to evaluate the attack performance with the Attack Success Rate (ASR) score, which measures the ratio of cases that are detected before the attack and undetected after the attack. The results are shown below:
>
> | Distance | Close | Long |
> | --- | --- | --- |
> | ASR | 53.1 % (51/96) | 15.8% (15/95) |
>
> The table shows that our BEVCA method achieves a relatively high ASR of 53.1% in close distance, demonstrating a significant threat to real-world scenarios. We have added all the relevant details of our physical experiment in our latest paper version.
>
> Adapting our generated camouflage to real-world data, such as nuScenes, is not currently feasible. Our generated camouflage is a 3D texture property. Together with a compatible 3D car model, we can render them to obtain 2D images. Although the nuScenes dataset contains a wide variety of vehicles, it does not provide corresponding 3D car models. Therefore, our work cannot directly be applied to nuScenes. The generation and evaluation of the physical camouflage workflow are typically done in two phases: the simulation phase and the real-world phase. In the simulation phase, we optimize the 3D adversarial textures by considering various physical settings (distances, angles, light conditions, etc) with gradient-based optimization to ensure their effectiveness and robustness. In the real-world phase, we apply the adversarial camouflage to the real object to obtain real-world data for evaluation.
>
> **W2: Only BEVFormer is studied, add other line of the BEV model such as BEVDepth.**
>
> We appreciate your recommendation to include BEVDepth as a representative BEV-based multi-view 3D detector in our experiments. However, since the physical experiment has consumed most of our time and efforts (building a multi-view data platform, printing and attaching adversarial camouflage to a real-size car, arranging a road test for data collection), we will incorporate BEVDepth into our experimental evaluation in a later version of our paper as soon as possible.

---

> ### Author Response · Authors · 2025-11-22
>
> **Q1: Is fine-tuning dataset enough to adapt the model to the simulation domain?**
>
> Adapting a model from one domain to another for the same task has been widely studied before [1][2][3] as a domain adaptation task. Since pretraining with a large realism dataset enables the model to learn the primary features of a car, this car-feature knowledge can be transferred to the simulation domain. We follow the common practice to fine-tune the model with a small-sized simulation dataset, ensuring the model achieves a reasonable level of performance in the simulation domain.
>
> In the table below, we present the performance of our selected models (AP for 3D detection and IoU for segmentation) in three scenarios: the pre-trained model on the realism dataset (nuScenes), the pre-trained model on the simulation dataset, and the fine-tuned model on the simulation dataset.
>
> | Models | pretrain→ realism | pretrain→ simulation | finetune→simluation |
> | --- | --- | --- | --- |
> | bevformer-base | 0.645 | 0.309 | 0.618 |
> | bevformer-small | 0.580 | 0.324 | 0.721 |
> | bevformer-tiny | 0.453 | 0.243 | 0.514 |
> | bevformer-base-seg-det | 0.642 | 0.321 | 0.750 |
> | bevformer-small-seg-det | 0.622 | 0.333 | 0.634 |
> | LSS | 0.340 | 0.209 | 0.509 |
> | CVT | 0.329 | 0.234 | 0.609 |
> | SinBEVT | 0.369 | 0.238 | 0.570 |
>
> The table shows that the detection performance of the pre-trained models on simulation data decreases compared to their performance on realism data. After fine-tuning, the models’ performance increases to a similar level as the pretrain models on realism data, and even higher performance in some cases. Therefore, we believe our fine-tune dataset is enough to adapt the model to the simulation domain.
>
> [1] Kouw, Wouter M., and Marco Loog. "An introduction to domain adaptation and transfer learning." *arXiv preprint arXiv:1812.11806* (2018).
>
> [2] Farahani, Abolfazl, et al. "A brief review of domain adaptation." *Advances in data science and information engineering: proceedings from ICDATA 2020 and IKE 2020* (2021): 877-894.
>
> [3] Csurka, Gabriela, ed. *Domain adaptation in computer vision applications*. Vol. 2. Cham: Springer International Publishing, 2017.
>
>
> **Q2: Optimizing L_det or L_bev + No Mask leads to almost no performance degradation in Table 3?**
>
> We started our adversarial texture optimization experiment by only using L_bev without masking, and we found that the attack performance of the generated texture is minimal. We then conduct a deeper analysis by visualizing the raw BEV difference feature, which is used to compute L_bev. The BEV difference feature is obtained by comparing the BEV feature of the target car with that of the car without the target car. Ideally, the high activation part of the BEV feature difference is only due to the existence of the target car. However, as shown in Figure 3, we also observe that some activation parts are scattered in random regions, which are not relevant to the target car. We suspect it is due to the randomness of the BEV model output. Moreover, we also found that the sum of these noisy activations is actually much higher than the activations relevant to the target car, as the target car only occupies a tiny part of the entire BEV perception area (around 3%). Hence, if we directly use the raw BEV difference feature to compute L_bev, which is mainly due to the randomness of the model output rather than the existence of the target car, it cannot generate an effective adversarial texture to evade the target car from the BEV model.
>
> Meanwhile, when we use L_det as the adversarial loss, the optimizing gradient needs to additionally backward through the detection decoder compared to the adversarial loss based on the BEV feature. This additional path makes L_det less direct and effective compared to the L_bev with mask loss. We have updated the above explanation in the 'Effectiveness of different adversarial losses' section of the ablation study in our latest paper version.
>
> **Q3: What models are used as the source model, and what is used as the target model?**
>
> As mentioned in the “target models” section of the experimental settings, BEVFormer-base is the only source (white-box) model to generate adversarial camouflage patterns. All the other models (other BEVFomer series for 3D detection task, LSS, CVT, and SinBEVT for segmentation) are used as black-box models to evaluate the transferability of our method.

---

> > ### Comment · Reviewer_UpWB · 2025-11-25
> >
> > I appreciate the authors effort to build a real-world data platform to validate the effectiveness of the proposed BEVCA on real-world data. I am willing to increase my rating, provided that no other significant concern is raised by other reviewers.

---

> > > ### Author Response · Authors · 2025-11-25
> > >
> > > Thank you very much for your kind and encouraging feedback! We sincerely appreciate your recognition of our efforts to conduct the physical world experiment to validate the effectiveness of BEVCA. We remain committed to addressing any remaining concerns raised by other reviewers and ensuring the robustness and clarity of our contributions.

---

### Author Response · Authors · 2025-12-04
**The rebuttal summary for AC (Part 1/2)**

Dear Area Chair,

Thank you for your valuable time and effort to review our work. To facilitate the review process, we would like to provide you with a summary of the key points we have made in response to reviewer feedback during the rebuttal period.

All reviewers acknowledged the importance of our framework for generating adversarial camouflage against multi-view BEV-based 3D perception models. **All the responding reviewers (Reviewers UpWB, BwNt, and anFR) have acknowledged that the physical world experiments have strengthened our proposed work**, addressing the main concerns regarding the simulation and realism gaps in our work. They also recognized the **solid attack performance of the generated camouflage** (Reviewers UpWB, BwNt, and 4tc6), the **paper’s novelty** (Reviewers UpWB and 4tc6), and **clear presentation of the paper** (Reviewer BwNt).

> ### **1. Physical Experiment Validation (Primary Contribution)**

**Original Concern:**

Due to the limited time and resources, we did not include the physical experiment to validate our camouflage method in the initial paper version. All four reviewers identify this as the paper's primary weakness.

**Our response:**

**We have successfully conducted** **full-scale physical-world experiments** **to validate BEVCA's effectiveness in real-world scenarios**, directly addressing the most significant concern raised by all four reviewers. We believe this experiment represents a substantial engineering contribution since it takes a substantial amount of workload to complete:

- **Custom Multi-View Data Collection Platform**: Built a six-camera sensor platform following nuScenes camera configurations (positions and angles) to ensure compatibility with multi-view perception models trained on nuScenes datasets.
- **Full-Size Vehicle Implementation**: Printed and applied our generated BEVCA camouflage (2D UV map) onto a real-life-size Audi vehicle.
- **Comprehensive Real-World Dataset**: Collected 191 samples per condition (normal painting vs. camouflage) across various azimuths and distances, split into close-range (<10m: 96 samples) and long-range (>10m: 95 samples) scenarios.
- **Significant Attack Performance**: Achieved **53.1% Attack Success Rate (ASR)** at close distances and 15.8% at long distances, demonstrating a significant real-world threat to real-world autonomous driving systems.

This physical validation bridged the critical sim2real gap that reviewers identified as the paper's primary weakness. We have added the physical experiment content to the latest version of our paper in the experiments and appendix sections. All the responding reviewers have acknowledged our effort to conduct the physical world experiments.

---

> ### Author Response · Authors · 2025-12-04
> **The rebuttal summary for AC (Part 2/2)**
>
> Continue from part one of the rebuttal summary.
>
> > ### **2. Novelty concerns**
>
> **Original concern:**
>
> Reviewers UpWB and 4tc6 express positive feedback regarding the novelty of the work, while Reviewers BwNt and anFR have concerns about the sufficiency of the BEV attack contribution compared to feature-level attack and its similarity to previous 3D texture attack methods.
>
> **Our response:**
>
> Our work addresses domain-specific challenges and introduces novel technical contributions that surpass previous feature-level and 3D texture attacks.
>
> - **BEV features with a delicate mask technique for effective and transferable attack:** Our insight is that most **high-performing multi-view 3D perception models rely on BEV features** for various downstream tasks. Therefore, by attacking BEV features, we can generate adversarial examples that transfer effectively across different models. However, direct use of BEV features for adversarial loss will introduce substantial noise from the irrelevant car-texture-sensitive regions, leading to ineffective attack. Therefore, we need to pinpoint the car-texture-sensitive BEV feature for vehicle texture optimization. To tackle this, we analyze the BEV difference features obtained from datasets with and without the target car and observe a strong correlation between the car-texture-sensitive BEV regions and the ground-truth car positions. Based on this observation, we **introduce a heuristic mask technique to accurately locate car-relevant BEV features**. We compare the attacks with other intermediate features, and the results show that **attacking the BEV feature with the mask technique achieves the most effective and transferable camouflage across diverse BEV-based perception models.**
>
> - **Multi-view neural rendering module: the proposed module enables accurate and consistent rendering of textured cars across multiple camera views in a multi-view dataset.** This module addresses a key engineering bottleneck of rendering the specific vehicle from different camera settings. To achieve this, we perform two coordinate transformations among three coordinate systems (Carla, nuScenes, and Pytorch3D), ensuring geometric alignment and view consistency. The proposed module enables gradient-based texture optimization against multi-view perception models. Moreover, it can serve as a general-purpose multi-view neural rendering tool for other vision and perception research tasks.
> - **Full-Size Physical Experiments:** We are the first work to provide full-size camouflage evaluation in the physical world, while the previous works only evaluated with a small-sized car model.
>
> We clarify that the novelty of our work lies in the BEV feature attack with the mask technique and the multi-view neural rendering module. The combination of these contributions allows us to achieve effective and transferable attacks against multi-view BEV-based 3D perception models.
>
> > ### **3. Practical use of our attack**
>
> **Original concern:**
>
> Reviewer anFR has concerns about how our attack can be concurrently used to enhance the safety of AD systems, and simple defense methods, such as adversarial patch detectors, may counter our attackers.
>
> **Our response:**
>
> We demonstrate that our generated adversarial examples can enhance the model's robustness with adversarial training. Additionally, we demonstrate that simple defenses, such as adversarial patch detectors, are ineffective in counteracting our attacks.
>
> - **Adversarial training to improve robustness:** We generate an adversarial training dataset to finetune the original BEV-based perception models. The original models experience a significant drop in performance when exposed to our adversarial camouflage. **After adversarial training, all models recover a substantial portion of their detection capability.** This shows that **our attack methodology is not only a means to expose weaknesses but also a practical tool for enhancing the safety and robustness of BEV-based perception models.**
> - **Adversarial patch detection experiments:** As suggested by reviewer anFR, we conducted experiments with NAPGuard and NutNet against our attack. We evaluated these defenses using our physical experiment datasets: the Normal-Paint Car dataset (clean) and the BEVCA Camouflaged Car dataset (adversarial). Our results indicate that **neither defense is effective in countering our 3D camouflage attack**. Specifically, both methods fail to accurately detect the adversarial texture due to a high false-negative rate.**
>
> These experiments demonstrate that current patch-based defenses are insufficient against our method.
>
> These clarifications demonstrate that BEVCA makes meaningful contributions to adversarial robustness in autonomous driving and provides a foundation for future work on multi-view 3D perception security.
>
> Thank you for your consideration.
>
> Sincerely, The Authors of the submission 8695

---

### Note · Authors · 2026-01-26

I have read and agree with the venue's withdrawal policy on behalf of myself and my co-authors.

---

### Meta-Review · Area_Chair_EFKq · 2026-01-07

**Summary:**

The reviewers generally agree that the paper addresses an important and timely problem—adversarial attacks on multi-view BEV-based 3D perception systems—and acknowledge the strong empirical attack performance, particularly in black-box and transfer settings. The primary concerns across reviews centered on (i) the lack of real-world validation in the initial submission, (ii) questions about the novelty of attacking BEV features relative to prior feature-level and 3D texture-based attacks, and (iii) the realism and generality of the experimental setup, including reliance on synthetic data and limited model coverage.

During rebuttal, the authors substantially addressed the most critical concern by adding full-scale physical-world experiments, which were widely acknowledged by reviewers as a significant strengthening of the paper and a meaningful contribution. While some reviewers continue to view the novelty as incremental—arguing that feature-level attacks are well studied—the authors clarified the domain-specific challenges of multi-view BEV-based perception and articulated clearer technical distinctions, particularly the BEV masking strategy and multi-view neural rendering module.

The AC read the paper again and does agree with the major comments raised by the reviewers.

**Reviewer Concerns:**

### concerns  addressed by the rebuttal

- The most significant concern raised by all reviewers—the lack of real-world validation and the sim-to-real gap—was convincingly addressed through the addition of full-scale physical experiments on a real vehicle with a multi-camera setup.
- Multiple reviewers explicitly acknowledged that these experiments substantially strengthened the paper and improved its practical relevance. The authors also clarified several experimental details, including the effectiveness of different adversarial losses, transferability evaluation protocols, and feature visualizations, which addressed earlier questions about attack design choices and interpretability.

### Concerns that remain outstanding:

- Some reviewers remain unconvinced about the level of conceptual novelty, noting that feature-level attacks are well established and questioning whether attacking BEV features constitutes a sufficiently strong standalone contribution. Although the authors provided detailed justification and ablation results, at least one reviewer continues to view the approach as a domain adaptation of existing techniques rather than a fundamentally new attack paradigm.

- In addition, concerns persist regarding the scope of applicability: the method is limited to BEV-based multi-view perception models, and its feasibility and realism in unconstrained real-world settings (e.g., legality, stealthiness, and deployment assumptions of full-vehicle camouflage) remain debated. These issues do not invalidate the technical contributions but reflect ongoing disagreement about generality and real-world threat modeling rather than experimental soundness.

**Reviewer Scores:**

Reviewer UpWB
Initial score: 4 (marginally below acceptance threshold).
After the rebuttal and the addition of full-scale physical-world experiments, the reviewer explicitly stated willingness to increase their rating, provided no new major concerns arose. Based on this response, the score would likely increase to 6 (marginally above acceptance threshold).

Reviewer BwNt
Initial score: 4 (marginally below acceptance threshold).
The reviewer acknowledged that the newly added real-world experiments strengthened the paper but remained unconvinced about the level of novelty, continuing to view the approach as fundamentally similar to prior 3D texture-based attacks. Given this position, the score would likely remain 4, or at most increase slightly to 5, but still reflect reservations.

Reviewer 4tc6
Initial score: 6 (marginally above acceptance threshold).
This reviewer viewed the core ideas favorably and primarily requested stronger real-world validation. The rebuttal directly addressed this concern with physical experiments, which aligns well with the reviewer’s stated expectations. The score would likely remain 6 or increase modestly to 7, reflecting increased confidence in real-world relevance.

Reviewer anFR
Initial score: 4 (marginally below acceptance threshold).
Despite extensive rebuttal, this reviewer explicitly stated that their concerns about novelty, scope, feasibility, and safety implications remained largely unaddressed. As such, their score would likely remain 4, with little indication of upward revision.

---

### Decision · Program_Chairs · 2026-01-26

Reject